# PROBTS: A UNIFIED TOOLKIT TO PROBE DEEP TIME-SERIES FORECASTING

## ABSTRACT

Time-series forecasting serves as a linchpin in a myriad of applications, spanning various domains. With the growth of deep learning, this arena has bifurcated into two salient branches: one focuses on crafting specific neural architectures tailored for time series, and the other harnesses advanced deep generative models for probabilistic forecasting. While both branches have made significant progress, their differences across data scenarios, methodological focuses, and decoding schemes pose profound, yet unexplored, research questions. To bridge this knowledge chasm, we introduce `ProbTS`, a pioneering toolkit developed to synergize and compare these two distinct branches. Endowed with a unified data module, a modularized model module, and a comprehensive evaluator module, `ProbTS` allows us to revisit and benchmark leading methods from both branches. The scrutiny with `ProbTS` highlights their distinct characteristics, relative strengths and weaknesses, and areas that need further exploration. Our analyses point to new avenues for research, aiming for more effective time-series forecasting.[1]

## 1 INTRODUCTION

Time-series forecasting serves as a cornerstone in many application scenarios across a plethora of domains, such as energy forecasting in solar applications (Rajagukguk et al., 2020), traffic prediction (Ghosh et al., 2009), climate projections (Angryk et al., 2020), and sustainable systems (Tai et al., 2023). Each scenario is unique, with distinct nuances in forecasting horizons, inherent temporal dependencies, and variations in data distributions.

With the meteoric rise of deep learning techniques in recent years (LeCun et al., 2015), deep time-series forecasting methods have increasingly gained traction. Diving deep into the existing methodologies, two salient research branches in this domain have emerged (Lim & Zohren, 2021). The first branch propelled by the groundbreaking success of customized neural network architectures in image and language domains, such as ResNets (He et al., 2016) and Transformer architectures (Vaswani et al., 2017). This branch focuses on crafting neural architectures tailored for time-series data representation (Oreshkin et al., 2020; Nie et al., 2023). On the other hand, the second branch is rooted in the advancements of deep generative models (Dinh et al., 2017; Papamakarios et al., 2017; Ho et al., 2020), looking to harness advanced probabilistic estimation methods for capturing intricate data distributions (Rasul et al., 2021b; Tashiro et al., 2021).

Given the diverse nature of time series forecasting tasks, these branches have pursued multiple avenues of innovation: from catering to different data characteristics and forecasting horizons to devising methodological focuses ranging from point forecasting via intricate neural architectures to probabilistic forecasting with nuanced density estimations. Furthermore, the branches exhibit distinct decoding schemes for multi-horizon forecasting, with the former majorly adopting non-autoregressive strategies and the latter inclining towards auto-regressive schemes.

Delving into the details, the first branch, with studies such as Zhou et al. (2021); Wu et al. (2021); Zhang et al. (2022); Challu et al. (2023); Nie et al. (2023), predominantly targets long-term forecasting, where data often reveals strong trends and seasonality patterns. Contrarily, the second branch, as showcased by works like (Salinas et al., 2019; Rasul et al., 2021b; Tashiro et al., 2021), is more

---

[1]This toolkit will be open-sourced.

oriented towards short-term forecasting, focusing on capturing detailed local variations and omitting significant trends or seasonality. Since different data characteristics and forecasting horizons may prefer widely different designs, this divergence naturally begs the question: how would the methodologies from one branch perform in scenarios traditionally addressed by the other?

From a methodological viewpoint, while the first branch specializes in neural network architecture design with inductive biases specifically for time-series data, they often restrict themselves to point forecasts. In contrast, the second branch, even with its keen interest in probabilistic forecasting, leans towards conventional neural network designs. The mystery then arises: what are the relative strengths and weaknesses of these two branches? And, can we integrate the strengths of both branches to revolutionize time-series forecasting?

Moreover, the decoding schemes chosen by these two branches exhibit a pronounced divergence. Methods in the first branch predominantly favor the non-autoregressive approach, projecting all future horizons in a single step. Whereas many methodologies in the second branch, illustrated by studies like (Salinas et al., 2019; Rasul et al., 2021b;a), adhere to the autoregressive scheme, generating forecasts step by step. This stark contrast between the two branches raises a compelling question: what underlying motivations and considerations steer this distinctive choice in decoding schemes? And, crucially, in the arena of time-series forecasting, which scheme showcases superior performance under diverse data scenarios?

Addressing these pressing research questions is paramount, as it can offer invaluable insights into the challenges and budding opportunities that come from harmonizing these two distinct research branches. Yet, despite the pressing need, there is an evident gap in the literature: no solution currently bridges the chasm between these branches, especially given the vast divergences in data scenarios, methodological focuses, and decoding schemes.

To enable comprehensive investigations on these inspiring research questions and to empower future research towards more effective time-series forecasting, this paper introduces `ProbTS`, an innovative toolkit tailored to unify and compare the research branch emphasizing neural architecture designs with the one focusing on advanced probabilistic estimations. At its core, `ProbTS` offers a unified *data* module, a modularized *model* module, and a holistic *evaluator* module. By harnessing `ProbTS`, we provide a comprehensive benchmark of some of the most pivotal methods from both branches, spanning a multitude of data scenarios and evaluation metrics. The outcomes of our analysis not only shed light on the previously stated research challenges but also unravel overlooked limitations in existing studies, charting the course for future time-series forecasting endeavors.

**Contributions.** We present `ProbTS`, a novel toolkit bridging two distinct time-series forecasting branches. It enables a thorough analysis and evaluation of top methodologies across various scenarios and metrics. Using `ProbTS`, we reveal insights into previously challenging questions:

- Our study discerns that long-term forecasting scenarios primarily exhibit stronger trends and seasonality, while short-term scenarios often manifest more complex data distributions. These differences in data characteristics profoundly influence the method designs of those two research branches.

- We recognize the strengths and shortcomings of various methodological focuses. Probabilistic forecasting methods excel in modeling complex data distributions, yet may yield subpar point forecasts despite superior distribution-level metrics. Conversely, customized network architectures, specifically tailored for trends and seasonality, display remarkable performance in long-term scenarios. However, their effectiveness in complex short-term forecasting scenarios remains unexplored. Furthermore, we observe that different decoding schemes have varying performances depending on specific data characteristics. These findings hint at a promising research potential, particularly in merging these two research branches across diverse data scenarios.

- We underscore the significance of a unified approach in both evaluation and learning. A singular focus on optimizing either probabilistic or non-probabilistic metrics may undermine the other's performance. This observation necessitates comprehensive evaluation metrics encompassing both distributional estimations and point-level forecasts, suggesting potential research into hybrid learning strategies that proficiently balance different measures.

Table 1: Comparison of two research branches in time-series forecasting. Comparison dimensions include forecasting horizons, forecasting paradigms, architecture designs, and decoding schemes.

| Studies | Fore. Horizon | | Fore. Paradigm | | Arch. Design | | Dec. Scheme | |
|---|---|---|---|---|---|---|---|---|
| | Short | Long | Point | Prob. | General | Customized | Auto. | Non-auto. |
| N-BEATS (Oreshkin et al., 2020) | ✗ | ✓ | ✓ | ✗ | ✗ | ✓ | ✗ | ✓ |
| Autoformer (Wu et al., 2021) | ✗ | ✓ | ✓ | ✗ | ✗ | ✓ | ✗ | ✓ |
| Informer (Zhou et al., 2021) | ✗ | ✓ | ✓ | ✗ | ✗ | ✓ | ✗ | ✓ |
| Pyraformer (Liu et al., 2022) | ✗ | ✓ | ✓ | ✗ | ✗ | ✓ | ✗ | ✓ |
| N-HiTS (Challu et al., 2023) | ✗ | ✓ | ✓ | ✗ | ✗ | ✓ | ✗ | ✓ |
| LTSF-Linear (Zeng et al., 2023) | ✗ | ✓ | ✓ | ✗ | ✗ | ✓ | ✗ | ✓ |
| PatchTST (Nie et al., 2023) | ✗ | ✓ | ✓ | ✗ | ✗ | ✓ | ✗ | ✓ |
| TimesNet (Wu et al., 2023) | ✓ | ✓ | ✓ | ✗ | ✗ | ✓ | ✗ | ✓ |
| DeepAR (Salinas et al., 2020) | ✓ | ✗ | ✗ | ✓ | ✓ | ✗ | ✓ | ✗ |
| GP-copula (Salinas et al., 2019) | ✓ | ✗ | ✗ | ✓ | ✓ | ✗ | ✓ | ✗ |
| LSTM NVP (Rasul et al., 2021b) | ✓ | ✗ | ✗ | ✓ | ✓ | ✗ | ✓ | ✗ |
| LSTM MAF (Rasul et al., 2021b) | ✓ | ✗ | ✗ | ✓ | ✓ | ✗ | ✓ | ✗ |
| Trans MAF (Rasul et al., 2021b) | ✓ | ✗ | ✗ | ✓ | ✓ | ✗ | ✓ | ✗ |
| TimeGrad (Rasul et al., 2021a) | ✓ | ✗ | ✗ | ✓ | ✓ | ✗ | ✓ | ✗ |
| CSDI (Tashiro et al., 2021) | ✓ | ✗ | ✗ | ✓ | ✗ | ✓ | ✗ | ✓ |
| SPD (Bilos et al., 2023) | ✓ | ✗ | ✗ | ✓ | ✓ | ✗ | ✗ | ✓ |
| `ProbTS` (Ours) | ✓ | ✓ | ✓ | ✓ | ✓ | ✓ | ✓ | ✓ |

## 2  RELATED WORK

Time-series forecasting has witnessed significant advancements in recent years, giving rise to two distinct research branches: one focused on effective network architecture designs, and the other dedicated to advancing probabilistic estimation. In Table 1, we present representative studies from these two branches and compare them with `ProbTS` across various aspects. In the following, we provide a comprehensive review of these two branches and discuss existing time-series forecasting toolkits, highlighting the unique features of `ProbTS`.

**Neural Architecture Designs in Time-series Forecasting.**  A substantial body of research has been dedicated to enhancing neural architecture designs for time-series forecasting. Notable examples include extensions of multi-layer perceptrons (Oreshkin et al., 2020; Fan et al., 2022; Zhang et al., 2022; Challu et al., 2023; Zeng et al., 2023; Ekambaram et al., 2023), customized recurrent or convolutional neural networks (Lai et al., 2018; LIU et al., 2022; Wu et al., 2023), and more recently, various Transformer-based variants developed (Vaswani et al., 2017; Li et al., 2019; Zhou et al., 2021; Wu et al., 2021; Zhou et al., 2022; Liu et al., 2022; Zhang & Yan, 2022; Cirstea et al., 2022; Nie et al., 2023). One common characteristic of studies in this branch is their typical use of the non-autoregressive decoding scheme, projecting into future horizons at once. Moreover, a significant portion of these studies, especially recent Transformer variants, primarily focuses on evaluating long-term forecasting scenarios, often characterized by prominent trending and seasonality patterns. Despite their advancements in neural architecture designs, these studies often limit themselves to point forecasts, capturing only the average variations of future values rather than the underlying data distribution. While approaches like quantile regression (Wen et al., 2017; Lim et al., 2021) can mitigate this limitation, they cannot replace the inherent capture of the data distribution.

**Probabilistic Estimation in Time-series Forecasting.**  In contrast to the first branch, which predominantly considers point forecasts, the second branch, commonly referred to as *deep probabilistic time-series forecasting*, aims to leverage deep neural networks to capture the complex data distribution of future time series. This branch has garnered broad research attention, ranging from early approaches employing pre-defined likelihood functions (Rangapuram et al., 2018; Salinas et al., 2020) or Gaussian copulas (Salinas et al., 2019; Drouin et al., 2022) to more recent methods harnessing advanced deep generative models (Rasul et al., 2021b;a; Tashiro et al., 2021; Bilos et al., 2023).

While studies in this branch can produce probabilistic forecasts, including distributional information about future time-series variations, they primarily evaluate short-term forecasting scenarios, inevitably overlooking the challenges posed by long-term trending or seasonality. A noteworthy observation is that, unlike studies in the first branch, which consistently prefer the non-autoregressive scheme, studies in this branch have employed both autoregressive (Rasul et al., 2021b;a) and non-autoregressive (Tashiro et al., 2021; Li et al., 2022) decoding schemes. Furthermore, in contrast to the flourishing development of specific neural architecture designs in the first branch, this research focus is relatively underexplored in the second branch. While a few methods (Tashiro et al., 2021; Li et al., 2022; Bergsma et al., 2022) incorporate customized neural architectures, more studies (de Bézenac et al., 2020; Rasul et al., 2021b;a; Bilos et al., 2023; Drouin et al., 2022) leverage standardized neural architectures to encode time-series representations.

**Time-series Forecasting Toolkits.** Parallel to the divergence of time-series research into two branches, we have also witnessed the emergence of two types of toolkits for time-series forecasting. The first type, primarily oriented towards point forecasting, includes Prophet (Taylor & Letham, 2018), sktime (Löning et al., 2019), tsai (Oguiza, 2022), TSlib (Wu et al., 2023), as well as other open-source implementations of studies from the first research branch. The second type, emphasizing probabilistic forecasting, encompasses GluonTS (Alexandrov et al., 2020) and PyTorchTS (Rasul et al., 2021b). However, these two types of toolkits, each specializing in a single forecasting paradigm, fall short of our research objective to unify the two distinct research branches. Two exceptions to this are PyTorchForecasting[2] and NeuralForecast[3], both of which incorporate customized neural architectures and probabilistic forecasters. Although these two toolkits come close to our work, they only cover a limited range of approaches, notably lacking more advanced probabilistic forecasting methods (Rasul et al., 2021b;a; Tashiro et al., 2021), and their pipelines are ill-suited for direct investigations into our research questions of interest. In contrast, `ProbTS` is purpose-built to unify and compare the latest research in neural architecture designs with cutting-edge studies in probabilistic forecasting. Most importantly, `ProbTS` aims to uncover overlooked limitations in existing approaches and identify new opportunities for future research to enhance the effectiveness of time-series forecasting.

## 3  THE PROBTS TOOLKIT

This section starts with a formal description of time series forecasting, then provides a concise overview of the ProbTS toolkit. To better illustrate the insights obtained by unifying customized neural architecture designs and advanced probabilistic estimations for deep time-series forecasting, we leave the detailed module descriptions to the Appendix A.

**Problem Formulation.** Formally, we denote an element of a multivariate time series as $x_t^k \in \mathbb{R}$, where $k$ represents the variate index and $t$ denotes the time index. At time step $t$, we have a multivariate vector $\boldsymbol{x}_t \in \mathbb{R}^K$. Each $x_t^k$ is associated with covariates $\boldsymbol{c}_t^k \in \mathbb{R}^N$, which encapsulates auxiliary information about the observations. Given a length-$T$ forecast horizon a length-$L$ observation history $\boldsymbol{x}_{t-L:t}$ and corresponding covariates $\boldsymbol{c}_{t-L:t}$, the objective in time series forecasting is to generate the vector of future values $\boldsymbol{x}_{t+1:t+T}$. In `ProbTS`, we decouple a model into an encoder $f_\phi$ and a forecaster $p_\theta$. An encoder is tasked with generating expressive hidden states $\boldsymbol{h} \in \mathbb{R}^D$. Under *autoregressive* decoding scheme, encoder forecasts variates using their past values, which can be formulated as $\boldsymbol{h}_t = f_\phi(\boldsymbol{x}_{t-1}, \boldsymbol{c}_t, \boldsymbol{h}_{t-1})$. Under the *non-autoregressive* scheme, the encoder generates all the forecasts in one step, which can be expressed as $\boldsymbol{h}_{t+1:t+T} = f_\phi(\boldsymbol{x}_{t-L:t}, \boldsymbol{c}_{t-L:t+T})$. A forecaster $p_\theta$ is employed either to directly estimate *point forecasts* as $\hat{\boldsymbol{x}}_t = p_\theta(\boldsymbol{h}_t)$, or to perform sampling based on the estimated *probabilistic distributions* as $\hat{\boldsymbol{x}}_t \sim p_\theta(\boldsymbol{x}_t|\boldsymbol{h}_t)$.

**An Overview of ProbTS.** The ProbTS toolkit comprises several integral modules designed to provide a comprehensive and fair evaluation. The *Data* module unifies diverse data scenarios, employs standardized pre-processing techniques. The *Model* module offers flexibility by accommodating various neural network architectures, forecasting paradigms, and decoding schemes, allowing

---

[2] https://github.com/jdb78/pytorch-forecasting
[3] https://github.com/Nixtla/neuralforecast

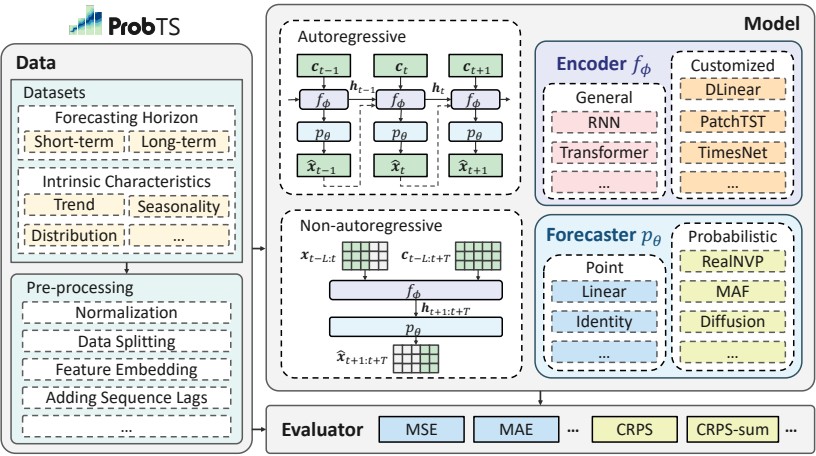

Figure 1: An overview of `ProbTS`.

the construction of diverse models. The *Evaluator* module integrates a diverse array of evaluation metrics, facilitating both point-level and distribution-level accuracy assessments. To maintain result integrity, we follow a standardized implementation process, including unified data splitting, standardization techniques, and fair hyperparameter tuning settings. Detailed information on each module and the experimental setup can be found in Appendix A and Appendix D, respectively.

## 4 PROBE DEEP TIME SERIES FORECASTING

In this section, we embark on a comparative study of deep time series forecasting methods, initiating with the selection of baseline models. We then provide an overview of our unique insights derived from ProbTS (Section 4.1), achieved through a collective comparison of different aspects of these models. Subsequently, we include detailed discussions on data scenarios (Section 4.2), different methodological focuses (Sections 4.3 and 4.4), decoding schemes (Section 4.5), and the efficiencies of these approaches (Section 4.6).

**Baseline Selection.** We incorporate a diverse range of models for our baseline analysis. For probabilistic approaches, we include models employing advanced density estimation techniques such as normalizing flows (GRU NVP, GRU MAF, and Trans MAF) and diffusion models (TimeGrad and CSDI). To assess the impact of probabilistic modeling, we consider general neural network architectures like Linear, GRU, and Transformer, coupled with a linear forecaster devoid of probabilistic estimation. We integrate global mean and batch mean as baselines to evaluate forecasting difficulty. Moreover, we incorporate TimesNet and PatchTST, models with architectures refined for time series, and N-HiTS and LTSF-Linear (NLinear and DLinear), recognized for their unique designs for modeling time series data.

### 4.1 INSIGHTS UNCOVERED WITH PROBTS: AN OVERVIEW

Utilizing the unified perspective afforded by `ProbTS`, we revisit two significant research branches in deep time-series forecasting and unveil critical insights.

**Advanced Probabilistic Methods: Achievements in Complex Data Distributions and Challenges in Long-term Forecasting.** Advanced probabilistic methods have shown their strengths in modeling complex data distributions. However, these methods encounter significant challenges when applied to long-term forecasting scenarios. Specifically, the complex data distributions in these scenarios are often masked by prominent trends and seasonality (Section 4.2), posing challenges for approaches focused solely on probabilistic modeling. These methods predominantly employ an auto-regressive decoding scheme, which is susceptible to substantial error propagation over extended horizons marked by prominent trends (Section 4.5). Additionally, the non-autoregressive

probabilistic method, CSDI, grapples with computational inefficiency and high memory requirements in long-term forecasting (Section 4.6). These issues, often overlooked in previous works, highlight the need for more comprehensive evaluations and further research into advanced probabilistic methods for long-term forecasting.

**Customized Neural Architecture Designs: Superiority in Long-term Forecasting and Unmet Challenges in Short-term Scenarios.** Customized neural architecture designs have demonstrated superiority in long-term forecasting scenarios, particularly those driven by prominent trends and seasonality (Section4.4). However, short-term forecasting scenarios, characterized by their complex data distributions, pose a different set of challenges that are not addressed by these architectures. Surprisingly, our comparative studies reveal that even without customized network architectures, autoregressive models can perform well in these scenarios, particularly when strong seasonality is present (Section4.5). This finding challenges the notion of disregarding autoregressive schemes in long-term contexts and underscores the potential for further research into customized neural architecture designs that can handle complex data distributions in short-term forecasting scenarios.

**Balancing Probabilistic and Non-probabilistic Metrics: A Unified Perspective.** Our study indicates that a focus on optimizing either probabilistic metrics (such as CRPS) or non-probabilistic metrics (such as NAME) may compromise the performance on the other (Section 4.3). This finding highlights the importance of this unified study in both evaluation and learning. It underscores the need for more comprehensive evaluation metrics that take into account both probabilistic scores and point-level forecasts. Furthermore, it suggests potential research directions towards the development of hybrid learning strategies that excel in both probabilistic and non-probabilistic metrics, meeting the practical needs of both accurate distributional estimations and precise point-level forecasts.

Table 2: Quantitative assessment of intrinsic characteristics for each dataset. To eliminate ambiguity, we use the suffix "-S" and "-L" to denote short-term and long-term forecasting datasets, respectively. The JS Div denotes Jensen–Shannon divergence, where a lower score indicates closer approximations to a Gaussian distribution.

| Dataset | Exchange-S | Solar-S | Electricity-S | Traffic-S | Wikipedia-S | ETTm1-L | ETTm2-L |
|---|---|---|---|---|---|---|---|
| Trend $F_T$ | 0.9982 | 0.1688 | 0.6443 | 0.2880 | 0.5253 | 0.9462 | 0.9770 |
| Seasonality $F_S$ | 0.1256 | 0.8592 | 0.8323 | 0.6656 | 0.2234 | 0.0105 | 0.0612 |
| JS Div. | 0.2967 | 0.5004 | 0.3579 | 0.2991 | 0.2751 | 0.0833 | 0.1701 |

| Dataset | ETTh1-L | ETTh2-L | Electricity-L | Traffic-L | Weather-L | Exchange-L | ILI-L |
|---|---|---|---|---|---|---|---|
| Trend $F_T$ | 0.7728 | 0.9412 | 0.6476 | 0.1632 | 0.9612 | 0.9978 | 0.5438 |
| Seasonality $F_S$ | 0.4772 | 0.3608 | 0.8344 | 0.6798 | 0.2657 | 0.1349 | 0.6075 |
| JS Div. | 0.0719 | 0.1422 | 0.1533 | 0.1378 | 0.1727 | 0.1082 | 0.1112 |

## 4.2 DATASET SCENARIOS

By unifying data scenarios, we can ascertain how existing methods perform on previously unevaluated data scenarios. This enables a deeper understanding of the motivations behind distinct designs, which are usually influenced by varied data characteristics.

**Differences in data characteristics.** Table 2 reveals that certain datasets, such as Exchange and ETT, exhibit robust trends with limited seasonality. On the other hand, datasets such as Solar and Traffic display reduced trendiness but substantial seasonality dominance. These variations impose distinct demands on a model's ability to handle trends and seasonality effectively. Moreover, the table illustrates the considerable divergences in data distributions among datasets. Typically, datasets closely adhering to a Gaussian distribution, such as Wikipedia-S, do not require much distribution modeling capabilities. Whereas datasets that deviate from the Gaussian distribution, e.g., Solar-S and Electricity-S, necessitate more advanced distribution modeling designs.

**Impacts of forecasting horizons.** To gain intuitive insights, we present visualizations of time series instances in Appendix B. Figure 2 and Figure 3 visually convey the impacts of varying forecasting horizons, showcasing instances of time series for short-term and long-term datasets respectively.

Table 3: Results (mean$_{std}$) on short-term forecasting datasets. We obtain mean and standard error metrics by re-training and evaluating five times.

| Model | Exchange Rate | | Solar | | Electricity | | Traffic | | Wikipedia | |
|---|---|---|---|---|---|---|---|---|---|---|
| | CRPS | NMAE | CRPS | NMAE | CRPS | NMAE | CRPS | NMAE | CRPS | NMAE |
| Glob. mean | 0.188 | 0.188 | 1.403 | 1.403 | 0.412 | 0.412 | 0.540 | 0.540 | 0.577 | 0.577 |
| Batch mean | 0.012 | 0.012 | 1.244 | 1.244 | 0.365 | 0.365 | 0.503 | 0.503 | 0.336 | 0.336 |
| Linear | $0.012_{001}$ | $0.012_{001}$ | $0.704_{036}$ | $0.704_{036}$ | $0.138_{009}$ | $0.138_{009}$ | $0.327_{032}$ | $0.327_{032}$ | $0.874_{151}$ | $0.874_{151}$ |
| GRU | $0.013_{002}$ | $0.013_{002}$ | $0.594_{144}$ | $0.594_{144}$ | $0.134_{009}$ | $0.134_{009}$ | $0.193_{002}$ | $0.193_{002}$ | $0.394_{013}$ | $0.394_{013}$ |
| Transformer | $0.016_{001}$ | $0.016_{001}$ | $0.538_{066}$ | $0.538_{066}$ | $0.115_{005}$ | $0.115_{005}$ | $0.204_{006}$ | $0.204_{006}$ | $0.408_{011}$ | $0.408_{011}$ |
| N-HiTS | $0.012_{003}$ | $0.012_{003}$ | $0.572_{020}$ | $0.572_{020}$ | $0.074_{003}$ | $0.074_{003}$ | $0.193_{002}$ | $0.193_{002}$ | $0.332_{011}$ | $0.332_{011}$ |
| NLinear | $\underline{0.010}_{000}$ | **$0.010_{000}$** | $0.560_{002}$ | $0.560_{002}$ | $0.083_{002}$ | $0.083_{002}$ | $0.233_{001}$ | $0.233_{001}$ | $0.321_{001}$ | $0.321_{001}$ |
| DLinear | $0.012_{001}$ | $0.012_{001}$ | $0.547_{009}$ | $0.547_{009}$ | $0.095_{006}$ | $0.095_{006}$ | $0.273_{012}$ | $0.273_{012}$ | $1.046_{037}$ | $1.046_{037}$ |
| PatchTST | $\underline{0.010}_{000}$ | **$0.010_{000}$** | $0.496_{002}$ | $0.496_{002}$ | $0.076_{001}$ | $0.076_{001}$ | $0.202_{001}$ | $0.202_{001}$ | $\underline{0.257}_{001}$ | **$0.257_{001}$** |
| TimesNet | $0.011_{001}$ | $0.011_{001}$ | $0.507_{019}$ | $0.507_{019}$ | $0.071_{002}$ | $0.071_{002}$ | $0.205_{002}$ | $0.205_{002}$ | $0.304_{002}$ | $0.304_{002}$ |
| GRU NVP | $0.016_{003}$ | $0.020_{003}$ | $0.396_{021}$ | $0.507_{022}$ | $0.055_{002}$ | $0.073_{002}$ | $0.161_{006}$ | $0.203_{009}$ | $0.282_{003}$ | $0.330_{003}$ |
| GRU MAF | $0.015_{001}$ | $0.020_{001}$ | $0.386_{026}$ | $0.492_{027}$ | $\underline{0.051}_{001}$ | $\underline{0.067}_{001}$ | $\underline{0.131}_{006}$ | $\underline{0.165}_{009}$ | $0.281_{004}$ | $0.337_{005}$ |
| Trans MAF | $0.011_{001}$ | $0.014_{001}$ | $0.400_{022}$ | $0.503_{022}$ | $0.054_{004}$ | $0.071_{005}$ | **$0.129_{004}$** | **$0.165_{006}$** | $0.289_{008}$ | $0.344_{008}$ |
| TimeGrad | $0.011_{001}$ | $0.014_{002}$ | **$0.359_{011}$** | **$0.445_{023}$** | $0.052_{001}$ | $\underline{0.067}_{001}$ | $0.164_{091}$ | $0.201_{115}$ | $0.272_{008}$ | $0.327_{011}$ |
| CSDI | **$0.008_{000}$** | $\underline{0.011}_{000}$ | $0.366_{005}$ | $0.484_{008}$ | **$0.050_{001}$** | **$0.065_{001}$** | $0.146_{012}$ | $0.176_{013}$ | **$0.219_{006}$** | $\underline{0.259}_{009}$ |

Table 4: Results (mean$_{std}$) on long-term forecasting datasets. The input sequence length is set to 36 for the ILI dataset and 96 for the others. We obtain mean and standard error metrics by re-training and evaluating three times. Due to the excessive time consumption of CSDI in high-dimensional scenarios, results are unavailable in partial long-term forecasting datasets.

| Model | pred len | DLinear | | PatchTST | | GRU NVP | | TimeGrad | | CSDI | |
|---|---|---|---|---|---|---|---|---|---|---|---|
| | | CRPS | NMAE | CRPS | NMAE | CRPS | NMAE | CRPS | NMAE | CRPS | NMAE |
| ETTm1 | 96 | $0.282_{002}$ | $0.282_{002}$ | $0.272_{001}$ | **$0.272_{001}$** | $0.383_{053}$ | $0.488_{058}$ | $0.522_{105}$ | $0.645_{129}$ | **$0.236_{006}$** | $0.308_{005}$ |
| | 192 | $0.309_{004}$ | $\underline{0.309}_{004}$ | $\underline{0.295}_{001}$ | **$0.295_{001}$** | $0.396_{030}$ | $0.514_{042}$ | $0.603_{092}$ | $0.748_{084}$ | **$0.291_{025}$** | $0.377_{026}$ |
| | 336 | $0.338_{008}$ | $\underline{0.338}_{008}$ | $\underline{0.323}_{001}$ | **$0.323_{001}$** | $0.486_{032}$ | $0.630_{029}$ | $0.601_{028}$ | $0.759_{015}$ | **$0.322_{033}$** | $0.419_{042}$ |
| | 720 | $\underline{0.387}_{006}$ | $\underline{0.387}_{006}$ | **$0.353_{001}$** | **$0.353_{001}$** | $0.546_{036}$ | $0.707_{050}$ | $0.621_{037}$ | $0.793_{034}$ | $0.448_{038}$ | $0.578_{051}$ |
| ETTm2 | 96 | $0.138_{000}$ | $0.138_{000}$ | $\underline{0.132}_{001}$ | **$0.132_{001}$** | $0.319_{044}$ | $0.413_{059}$ | $0.427_{042}$ | $0.525_{047}$ | **$0.115_{009}$** | $0.146_{012}$ |
| | 192 | $0.163_{003}$ | $0.163_{003}$ | $\underline{0.157}_{001}$ | **$0.157_{001}$** | $0.326_{025}$ | $0.427_{033}$ | $0.424_{061}$ | $0.530_{060}$ | **$0.147_{008}$** | $0.189_{012}$ |
| | 336 | $0.188_{001}$ | $0.188_{001}$ | $0.176_{002}$ | **$0.176_{002}$** | $0.449_{145}$ | $0.580_{169}$ | $0.469_{049}$ | $0.566_{047}$ | $0.190_{018}$ | $0.248_{024}$ |
| | 720 | $\underline{0.219}_{003}$ | $\underline{0.219}_{003}$ | **$0.205_{001}$** | **$0.205_{001}$** | $0.561_{273}$ | $0.749_{385}$ | $0.470_{054}$ | $0.561_{044}$ | $0.239_{035}$ | $0.306_{040}$ |
| ETTh1 | 96 | $0.352_{011}$ | $0.352_{011}$ | **$0.328_{003}$** | **$0.328_{003}$** | $0.379_{030}$ | $0.481_{037}$ | $0.455_{046}$ | $0.585_{058}$ | $0.437_{018}$ | $0.557_{022}$ |
| | 192 | $0.393_{001}$ | $0.393_{001}$ | **$0.359_{002}$** | **$0.359_{002}$** | $0.425_{019}$ | $0.531_{018}$ | $0.516_{038}$ | $0.680_{058}$ | $0.496_{051}$ | $0.625_{065}$ |
| | 336 | $\underline{0.419}_{007}$ | $\underline{0.419}_{007}$ | **$0.384_{002}$** | **$0.384_{002}$** | $0.458_{054}$ | $0.580_{064}$ | $0.512_{026}$ | $0.666_{047}$ | $0.454_{025}$ | $0.574_{026}$ |
| | 720 | $\underline{0.502}_{029}$ | $\underline{0.502}_{029}$ | **$0.397_{002}$** | **$0.397_{002}$** | $0.502_{039}$ | $0.643_{046}$ | $0.523_{027}$ | $0.672_{015}$ | $0.528_{012}$ | $0.657_{014}$ |
| ETTh2 | 96 | $0.211_{027}$ | $0.211_{027}$ | $0.177_{000}$ | **$0.177_{000}$** | $0.432_{141}$ | $0.548_{158}$ | $0.358_{026}$ | $0.448_{031}$ | **$0.164_{013}$** | $0.214_{018}$ |
| | 192 | $0.238_{028}$ | $\underline{0.238}_{028}$ | **$0.201_{001}$** | **$0.201_{001}$** | $0.625_{170}$ | $0.766_{223}$ | $0.457_{081}$ | $0.575_{089}$ | $\underline{0.226}_{018}$ | $0.294_{027}$ |
| | 336 | $0.284_{008}$ | $\underline{0.284}_{008}$ | **$0.240_{001}$** | **$0.240_{001}$** | $0.793_{319}$ | $0.942_{408}$ | $0.481_{078}$ | $0.606_{095}$ | $\underline{0.274}_{022}$ | $0.353_{028}$ |
| | 720 | $0.307_{000}$ | $\underline{0.307}_{000}$ | **$0.252_{000}$** | **$0.252_{000}$** | $0.539_{090}$ | $0.688_{161}$ | $0.445_{016}$ | $0.550_{018}$ | $\underline{0.302}_{040}$ | $0.382_{030}$ |
| Electricity | 96 | $0.090_{001}$ | $0.090_{001}$ | **$0.086_{001}$** | **$0.086_{001}$** | $0.094_{003}$ | $0.118_{003}$ | $0.096_{002}$ | $0.119_{003}$ | $0.153_{137}$ | $0.203_{189}$ |
| | 192 | $\underline{0.095}_{001}$ | $\underline{0.095}_{001}$ | **$0.092_{001}$** | **$0.092_{001}$** | $0.097_{002}$ | $0.121_{003}$ | $0.100_{004}$ | $0.124_{005}$ | $0.200_{094}$ | $0.264_{129}$ |
| | 336 | $\underline{0.104}_{000}$ | $\underline{0.104}_{000}$ | $0.100_{000}$ | **$0.100_{000}$** | **$0.099_{001}$** | $0.123_{001}$ | $0.102_{007}$ | $0.126_{008}$ | - | - |
| | 720 | $0.122_{001}$ | $\underline{0.122}_{001}$ | $0.116_{000}$ | **$0.116_{000}$** | $\underline{0.114}_{013}$ | $0.144_{017}$ | **$0.108_{003}$** | $0.134_{004}$ | - | - |
| Traffic | 96 | $0.356_{009}$ | $0.356_{009}$ | $0.248_{001}$ | $0.248_{001}$ | **$0.187_{002}$** | **$0.231_{003}$** | $\underline{0.202}_{004}$ | $\underline{0.234}_{006}$ | - | - |
| | 192 | $0.346_{009}$ | $0.346_{009}$ | $0.245_{001}$ | $0.245_{001}$ | **$0.192_{001}$** | **$0.236_{002}$** | $\underline{0.208}_{003}$ | $\underline{0.239}_{004}$ | - | - |
| | 336 | $0.350_{008}$ | $0.350_{008}$ | $0.257_{002}$ | $0.257_{002}$ | **$0.201_{004}$** | $0.248_{006}$ | $\underline{0.213}_{003}$ | **$0.246_{003}$** | - | - |
| | 720 | $0.365_{009}$ | $0.365_{009}$ | $0.266_{001}$ | $0.266_{001}$ | **$0.211_{004}$** | $\underline{0.264}_{006}$ | $\underline{0.220}_{002}$ | **$0.263_{001}$** | - | - |
| Weather | 96 | $0.112_{001}$ | $0.112_{001}$ | $0.087_{002}$ | **$0.087_{002}$** | $0.116_{013}$ | $0.145_{017}$ | $0.130_{017}$ | $0.164_{023}$ | **$0.068_{008}$** | $0.087_{012}$ |
| | 192 | $0.122_{001}$ | $0.122_{001}$ | $\underline{0.090}_{001}$ | $\underline{0.090}_{001}$ | $0.122_{021}$ | $0.147_{025}$ | $0.127_{019}$ | $0.158_{024}$ | **$0.068_{006}$** | **$0.086_{007}$** |
| | 336 | $0.130_{002}$ | $0.130_{002}$ | $\underline{0.092}_{002}$ | **$0.092_{002}$** | $0.128_{011}$ | $0.160_{012}$ | $0.130_{006}$ | $0.162_{006}$ | **$0.083_{002}$** | **$0.098_{002}$** |
| | 720 | $0.144_{001}$ | $0.144_{001}$ | $\underline{0.094}_{001}$ | **$0.094_{001}$** | $0.110_{004}$ | $0.135_{008}$ | $0.113_{011}$ | $0.136_{020}$ | **$0.087_{003}$** | $\underline{0.102}_{005}$ |
| Exchange | 96 | $0.024_{000}$ | $0.024_{000}$ | **$0.023_{000}$** | **$0.023_{000}$** | $0.071_{006}$ | $0.091_{009}$ | $0.068_{003}$ | $0.079_{002}$ | $0.028_{003}$ | $0.036_{005}$ |
| | 192 | $\underline{0.035}_{000}$ | $\underline{0.035}_{000}$ | **$0.034_{000}$** | **$0.034_{000}$** | $0.068_{004}$ | $0.087_{005}$ | $0.087_{013}$ | $0.100_{019}$ | $0.045_{003}$ | $0.058_{005}$ |
| | 336 | $\underline{0.048}_{001}$ | $\underline{0.048}_{001}$ | **$0.048_{000}$** | **$0.048_{000}$** | $0.072_{002}$ | $0.091_{002}$ | $0.074_{009}$ | $0.086_{008}$ | $0.060_{004}$ | $0.076_{006}$ |
| | 720 | $0.075_{002}$ | $0.075_{002}$ | **$0.072_{000}$** | **$0.072_{000}$** | $0.079_{009}$ | $0.103_{009}$ | $0.099_{015}$ | $0.113_{016}$ | $0.143_{020}$ | $0.173_{020}$ |
| ILI | 24 | $0.213_{038}$ | $0.213_{038}$ | **$0.169_{005}$** | **$0.169_{005}$** | $0.257_{003}$ | $0.283_{001}$ | $0.275_{047}$ | $0.296_{044}$ | $0.250_{013}$ | $0.263_{012}$ |
| | 36 | $\underline{0.230}_{015}$ | $\underline{0.230}_{015}$ | **$0.156_{005}$** | **$0.156_{005}$** | $0.281_{004}$ | $0.307_{007}$ | $0.272_{057}$ | $0.298_{048}$ | $0.285_{010}$ | $0.298_{011}$ |
| | 48 | $\underline{0.221}_{009}$ | $\underline{0.221}_{009}$ | **$0.156_{008}$** | **$0.156_{008}$** | $0.288_{008}$ | $0.314_{009}$ | $0.295_{033}$ | $0.320_{025}$ | $0.285_{036}$ | $0.301_{034}$ |
| | 60 | $\underline{0.230}_{013}$ | $\underline{0.230}_{013}$ | **$0.147_{003}$** | **$0.147_{003}$** | $0.307_{005}$ | $0.333_{005}$ | $0.295_{083}$ | $0.325_{068}$ | $0.283_{012}$ | $0.299_{013}$ |

Figure 2 emphasizes that short-term contexts are primarily characterized by local variations. Conversely, Figure 3 demonstrates an augmented presence of seasonality and trends in datasets like Traffic, Electricity, and ETT under elongated forecasting horizons, enhancing predictability. This

shift highlights the importance of capturing local dynamics in short-term forecasting, while longer forecasting horizons necessitate models adept at modeling extended seasonality and trends.

## 4.3 POINT VS. PROBABILISTIC ESTIMATION

Beyond differing data scenarios, varied methodological focuses raise essential questions about the unique benefits of specialized model designs, especially probabilistic estimation.

**Probabilistic methods excel in modeling complex data distributions.** From Table 3, it is evident that probabilistic estimation methods, especially TimeGrad and CSDI, demonstrate superior performance in both NMAE and CRPS metrics in short-term forecasting. This advantage is particularly notable in the Solar, Electricity, and Traffic datasets, which show intricate data distributions in Table 2. This underscores the aptitude of probabilistic methods in addressing complex data distributions. Whereas for long-term forecasts, we cannot compare probabilistic and point estimation methods solely based on data distribution since the performance is multifacetedly affected.

**Probabilistic methods may produce poor point forecasts even with a superior CRPS score.** Table 4 illustrates that while probabilistic methods like CSDI demonstrate prowess in the CRPS metric, they lag in NMAE, reflecting precise distribution approximations but weaker point forecasts. This discrepancy is indicative of inherent limitations in current probabilistic models, shedding light on the prevailing preference for CRPS in evaluations. It highlights substantial opportunities for refining probabilistic approaches and underscores the critical need for holistic evaluation perspectives to enhance the reliability and precision of time-series forecasting.

The complex local variations inherent to shorter forecasting horizons, as detailed in Section 4.2, explain why most probabilistic forecasting studies opt for short-term datasets for model evaluation. However, proficiency in one aspect is insufficient in real-world applications, and the ability to handle other intrinsic data characteristics is equally crucial, which calls for a more comprehensive evaluation setting.

## 4.4 CUSTOMIZED VS. GENERAL NEURAL ARCHITECTURE

In addition to exploring the advantages of probabilistic estimation, the pressing question of whether a customized neural architecture with time-series inductive bias is essential remains unresolved.

**Customized network architecture performs remarkably on long-term forecasting.** Table 4 illustrates that in long-term forecasting, models with time-series inductive bias like DLinear and PatchTST significantly outperform models that employ advanced probabilistic estimation techniques but rely solely on general architectures. Given that data exhibit more pronounced seasonality and trends in longer forecast horizons, as discussed in Section 4.2, we attribute this superiority to their trend-seasonality decomposition techniques. This suggests that a general architecture is not enough for time-series modeling, and incorporating more unique designs tailored for time-series domain knowledge is expected.

**Customized network architecture on short-term forecasting remains underexplored.** Different from long-term forecasting, Table 3 suggests short-term forecasting does not significantly benefit from existing customized network architectures compared to general ones. Exceptions like the Exchange and Wikipedia datasets are influenced by their intrinsic characteristics; the former is smooth, allowing even basic models like batch mean to succeed (see Table 4.2), while the latter includes abundant outliers pose challenges for all models (see Table 7). Thus, the development and refinement of network architectures to better apprehend short-term fluctuations represent a pivotal and unmet challenge.

Our findings expose the performance gap of distinct designs across diverse data scenarios, explaining the preference of different methodological focuses for unique dataset settings. This insight uncovers a potential path to revolutionize time-series forecasting by harnessing the strengths of both neural network architectures with time-series inductive bias and probabilistic estimation methods. Such an amalgamation enables effective modeling of intricate data distributions while adeptly managing the inherent periodicity and seasonality of time series.

### 4.5 AUTOREGRESSIVE VS. NON-AUTOREGRESSIVE DECODING SCHEME

Since the decoding scheme is another critical aspect influencing time series forecasting performance, this section is dedicated to elucidating the strengths and weaknesses of both decoding methods and discussing how we should choose between them in different data scenarios.

**Autoregressive models excel with strong seasonality but struggle with pronounced trends.** Table 4 indicates that autoregressive methods generally exhibit inferior performance to non-autoregressive ones in long-term datasets. However, notable exceptions exist, autoregressive models display superior efficacy on the Traffic dataset, which is of minimal trend strength (see Table 2). Under this situation, the autoregressive methods even surpass the state-of-the-art non-autoregressive models. Detailed examination reveals the difficulties autoregressive models face in datasets with significant trends like ETT, while they fare well in modeling strong seasonality, as evident in the Electricity and Traffic datasets. This manifests the proficiency of autoregressive models in capturing seasonal patterns but their inadequacy in modeling long-term trends.

**Both decoding schemes perform equally well in short-term forecasting.** Table 3 implies a comparable performance between two decoding schemes in short-term forecasting datasets. This is possibly attributed to the limited impact of error propagation on autoregressive methods in short-term scenarios. Additionally, the flexibility of the autoregressive approaches makes it a feasible choice for accommodating diverse temporal structures and complexities in short-term prediction datasets.

The experimental findings reveal that while autoregressive methods are proficient in modeling seasonality, they falter in managing long-term trends, an area where non-autoregressive methods demonstrate competency. This divergence in expertise serves as a guideline for selecting decoding schemes for particular data scenarios. It also explains the preferential biases of the aforementioned two branches towards specific decoding manner. Interestingly, there are cases where autoregressive methods outperform non-autoregressive ones in long-term situations, especially in datasets with strong seasonal patterns. This finding leads us to speculate that enhancing autoregressive models to curb error propagation and adeptly handle trending data could make them strong challengers in providing long-term forecasting solutions.

### 4.6 COMPUTATIONAL EFFICIENCY

The necessity to depict intricate, high-dimensional data distributions in modern applications introduces considerable computational challenges in time-series forecasting, a crucial aspect often neglected by prior research. Thus, we provide a computational efficiency analysis here.

**Modeling high-dimensional probabilistic distributions demands substantial memory and is time-intensive.** In Figure 6a, a significant escalation in GPU memory consumption is observed with the increase in the number of variates, where CSDI—integrating non-autoregressive decoding schemes with a diffusion model—exhibits a particularly pronounced surge. Additionally, Figure 6b reveals that methods centered on probabilistic estimation entail longer inference times, especially the Diffusion model. These findings underscore the crucial need for optimizing memory efficiency and reducing inference time, particularly as the prediction horizon extends.

## 5 CONCLUSION

In conclusion, this paper has presented `ProbTS`, a novel toolkit developed to advance the field of time-series forecasting by synergizing and comparing research branches emphasizing neural architecture designs with the one focusing on advanced probabilistic estimations. Through `ProbTS`, we have answered several pivotal research questions stemming from the divergences in data scenarios, methodologies, and decoding schemes between these distinct branches. Looking ahead, there is immense potential in amalgamating the strengths of both branches to redefine the future of time-series forecasting. We anticipate that the `ProbTS` toolkit will act as a catalyst, expediting groundbreaking research in this domain, and unlocking new possibilities for refined and robust forecasting models.

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

## A  Components of the ProbTS Toolkit

**Data.**  The data module unifies varied data scenarios to facilitate thorough evaluation and implements standardized pre-processing techniques to ensure fair comparison. Moreover, we utilize a quantitative approach to visually delineate datasets' intrinsic characteristics, which employs decomposition to assess trends and seasonality in a time series and evaluate the similarity between data distribution and a Gaussian to depict the complexity of data distribution. Descriptions and statistics for each dataset are listed in Table 5, and a quantitative evaluation of their inherent properties is provided in Table 2. We attach the detailed quantitative calculation process in Appendix B.

**Model.**  The modularized model module accommodates diverse neural network architectures, forecasting paradigms, and decoding schemes. Adhering to the decoupled model formulation from Section 3, it enables the construction of various models by configuring the encoder $f_\phi$ and forecaster $p_\theta$. For example, point estimation methods like DLinear centralize their design in the encoder, using a linear layer or identity mapping as the forecaster, with non-autoregressive decoding. In contrast, probabilistic models like TimeGrad incorporate general neural architectures in the encoder and advanced probabilistic techniques in the forecaster, employing autoregressive decoding.

**Evaluator.**  The evaluator module integrates a diverse array of evaluation metrics such as Mean Absolute Error (MAE), Normalized Mean Absolute Error (NMAE), Mean Square Error (MSE), and Continuous Ranked Probability Score (CRPS), allowing for assessment of both point-level and distribution-level accuracies. We employ the NMAE metric for point-level evaluation to accommodate different scales of errors, and unlike previous studies (Rasul et al., 2021a; Tashiro et al., 2021) that used the $\text{CRPS}_{\text{sum}}$ metric, we utilize CRPS for our analysis for a refined evaluation of each variate's probability distribution accuracy. A detailed list of evaluation metrics and their formal definitions can be found in Appendix C.

**Implementation.**  To ensure the integrity of the results, `ProbTS` adheres to a standard implementation process, employing unified data splitting, standardization techniques, and adopting fair settings for hyperparameter tuning across all methods. We utilize reported optimal hyperparameters for models directly associated with specific datasets and conduct an extensive grid search to identify the most effective settings for those hyperparameters that were not available. Details regarding the experimental setup can be found in Appendix D.

Table 5: Dataset Summary.

| Horizon | Dataset | #var. | range | freq. | timesteps | Description |
|---|---|---|---|---|---|---|
| | ETTh1/h2 | 7 | $\mathbb{R}^+$ | H | 17,420 | Electricity transformer temperature per hour |
| | ETTm1/m2 | 7 | $\mathbb{R}^+$ | 15min | 69,680 | Electricity transformer temperature every 15 min |
| | Electricity | 321 | $\mathbb{R}^+$ | H | 26,304 | Electricity consumption (Kwh) |
| **Long-term** | Traffic | 862 | (0,1) | H | 17,544 | Road occupancy rates |
| | Exchange | 8 | $\mathbb{R}^+$ | Busi. Day | 7,588 | Daily exchange rates of 8 countries |
| | ILI | 7 | (0,1) | W | 966 | Ratio of patients seen with influenza-like illness |
| | Weather | 21 | $\mathbb{R}^+$ | 10min | 52,696 | Local climatological data |
| | Exchange | 8 | $\mathbb{R}^+$ | Busi. Day | 6,071 | Daily exchange rates of 8 countries |
| | Solar | 137 | $\mathbb{R}^+$ | H | 7,009 | Solar power production records |
| **Short-term** | Electricity | 370 | $\mathbb{R}^+$ | H | 5,833 | Electricity consumption |
| | Traffic | 963 | (0,1) | H | 4,001 | Road occupancy rates |
| | Wikipedia | 2,000 | $\mathbb{N}$ | D | 792 | Page views of 2000 Wikipedia pages |

## B  Quantifying the Characteristics of Datasets

**Trend & Seasonality**  To gain deeper insights into the dataset characteristics, we conducted a quantitative evaluation of trend and seasonality for each dataset, drawing upon methodologies out-

lined in the work of Wang et al. (2006). In particular, we employed a time series decomposition model expressed as:

$$y_t = T_t + S_t + R_t,$$

where $T_t$ represents the smoothed trend component, $S_t$ signifies the seasonal component, and $R_t$ denotes the remainder component. In order to obtain each component, we followed the STL decomposition approach [4].

In the case of strongly trended data, the variation within the seasonally adjusted data should considerably exceed that of the remainder component. Consequently, the ratio $\mathrm{Var}(R_t)/\mathrm{Var}(T_t + R_t)$ is expected to be relatively small. As such, the measure of trend strength can be formulated as:

$$F_T = \max\left(0, 1 - \frac{\mathrm{Var}(R_t)}{\mathrm{Var}(T_t + R_t)}\right).$$

The quantified trend strength, ranging from 0 to 1, characterizes the degree of trend presence. Similarly, the evaluation of seasonal intensity employs the detrended data:

$$F_S = \max\left(0, 1 - \frac{\mathrm{Var}(R_t)}{\mathrm{Var}(S_t + R_t)}\right).$$

A series with $F_S$ near 0 indicates minimal seasonality, while strong seasonality is indicated by $F_S$ approaching 1 due to the considerably smaller variance of $\mathrm{Var}(R_t)$ in comparison to $\mathrm{Var}(S_t + R_t)$.

Tables 6 depict the results for each dataset. Notably, the ETT datasets and the Exchange dataset manifest conspicuous trends, whereas the Electricity, Solar, and Traffic datasets showcase marked seasonality. Additionally, the Exchange dataset stands out with distinctive features. Figure 3 illustrates that with shorter prediction windows, the Exchange dataset sustains comparatively minor fluctuations, almost forming a linear trajectory. This enables effective forecasting through a straightforward batch mean approach. As the forecasting horizon extends, the dataset appears a more pronounced trend while retaining minimal seasonality.

**Data Distribution** To analyze the influence of data distribution on model performance, we measured the similarity between each dataset's distribution and the Gaussian distribution. Specifically, we computed the Jensen–Shannon divergence (Nielsen, 2019) within a fixed-length sliding window for each variate. A window size of 30 was used for short-term datasets and 336 for long-term ones. The average of these calculations yielded the overall degree of conformity of each dataset to the Gaussian distribution. These results are summarized in Table 6.

Table 6: Quantitative assessment of intrinsic characteristics for each dataset. To eliminate ambiguity, we use the suffix "-S" and "-L" to denote short-term and long-term forecasting datasets, respectively. The JS Div denotes Jensen–Shannon divergence, where a lower score indicates closer approximations to a Gaussian distribution.

| Dataset | Exchange-S | Solar-S | Electricity-S | Traffic-S | Wikipedia-S | ETTm1-L | ETTm2-L |
|---|---|---|---|---|---|---|---|
| Trend $F_T$ | 0.9982 | 0.1688 | 0.6443 | 0.2880 | 0.5253 | 0.9462 | 0.9770 |
| Seasonality $F_S$ | 0.1256 | 0.8592 | 0.8323 | 0.6656 | 0.2234 | 0.0105 | 0.0612 |
| JS Div. | 0.2967 | 0.5004 | 0.3579 | 0.2991 | 0.2751 | 0.0833 | 0.1701 |

| Dataset | ETTh1-L | ETTh2-L | Electricity-L | Traffic-L | Weather-L | Exchange-L | ILI-L |
|---|---|---|---|---|---|---|---|
| Trend $F_T$ | 0.7728 | 0.9412 | 0.6476 | 0.1632 | 0.9612 | 0.9978 | 0.5438 |
| Seasonality $F_S$ | 0.4772 | 0.3608 | 0.8344 | 0.6798 | 0.2657 | 0.1349 | 0.6075 |
| JS Div. | 0.0719 | 0.1422 | 0.1533 | 0.1378 | 0.1727 | 0.1082 | 0.1112 |

**Outliers** Outliers are data points that are significantly distant from the rest, which pose challenges in forecasting. We quantified outlier ratios from both global and local perspectives. The global view treats the entire dataset as a Gaussian distribution and identifies Z-score normalized values more than 3 standard deviations from the mean as outliers. The local perspective assesses outliers

---

[4]https://otexts.com/fpp2/stl.html

within a sliding window, following the same criterion. For short-term datasets, a window size of 30 is employed, while for long-term forecasting datasets, the window size is set to 336. We present the ratio of outliers in Table 7 for reference. From Table 7, we find that some datasets, such as Wikipedia-S, possess a high local ratio of outliers, which can have a large impact on short-term forecasting.

Table 7: Ratio of outliers (%). The suffix "-S" denotes short-term forecasting datasets, while "-L" signifies long-term forecasting datasets.

| Dataset | Exchange-S | Solar-S | Electricity-S | Traffic-S | Wikipedia-S | ETTm1-L | ETTm2-L |
|---|---|---|---|---|---|---|---|
| Local | 0.1718 | 0.2228 | 0.1333 | 0.6595 | 1.5435 | 0.4126 | 0.4231 |
| Global | 0.0871 | 0.0002 | 0.4210 | 1.6890 | 1.1758 | 1.1079 | 1.8764 |

| Dataset | ETTh1-L | ETTh2-L | Electricity-L | Traffic-L | Weather-L | Exchange-L | ILI-L |
|---|---|---|---|---|---|---|---|
| Local | 0.4937 | 0.4707 | 0.1529 | 1.4352 | 0.5106 | 0.2021 | 1.2422 |
| Global | 1.2951 | 2.1929 | 0.4134 | 1.5885 | 0.8323 | 0.0066 | 1.5735 |

**Data Visualization**   To offer a clearer insight into the characteristics of each dataset and the influence of varying forecasting horizons, we have illustrated instances of both short-term and long-term forecasting datasets in Figure 2 and Figure 3 respectively. Figure 2 reveals that in short-term scenarios, time series are primarily governed by local variations. On the other hand, as depicted in Figure 3, datasets like Traffic, Electricity, and ETT, under extended forecasting horizons, display enhanced seasonality and trends, making these series more predictable.

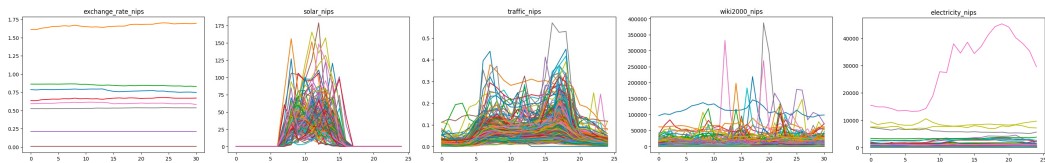

Figure 2: Time series samples extracted from the short-term forecasting dataset. The range of the x-axis is the pre-defined length of the prediction window in each dataset.

## C   EVALUATION METRICS

The `ProbTS` toolkit incorporates a comprehensive range of metrics, spanning both point-level and distribution-level, to offer a nuanced and multifaceted evaluation of forecasting models.

### C.1   POINT-LEVEL METRICS

For point-level metrics, we primarily focused on several measures that are predominantly used in the branch devoted to optimizing neural network architecture design.

**Mean Absolute Error (MAE)**   The Mean Absolute Error (MAE) quantifies the average absolute deviation between the forecasts and the true values. Since it averages the absolute errors, MAE is robust to outliers. Its mathematical formula is given by:

$$\text{MAE} = \frac{1}{K \times T} \sum_{i=1}^{K} \sum_{t=1}^{T} |x_{i,t} - \hat{x}_{i,t}|,$$

where $K$ is the number of variates, $L$ is the length of series, $x_{i,t}$ and $\hat{x}_{i,t}$ denotes the ground-truth value and the predicted value, respectively. For multivariate time series, we also provide the aggregated version:

$$\text{MAE}_{\text{sum}} = \frac{1}{T} \sum_{t=1}^{T} |x_t^{\text{sum}} - \hat{x}_t^{\text{sum}}|,$$

where $x_t^{\text{sum}}$ and $\hat{x}_t^{\text{sum}}$ are the summation across the dimension $K$ of $x_{i,t}$ and $\hat{x}_{i,t}$, respectively.

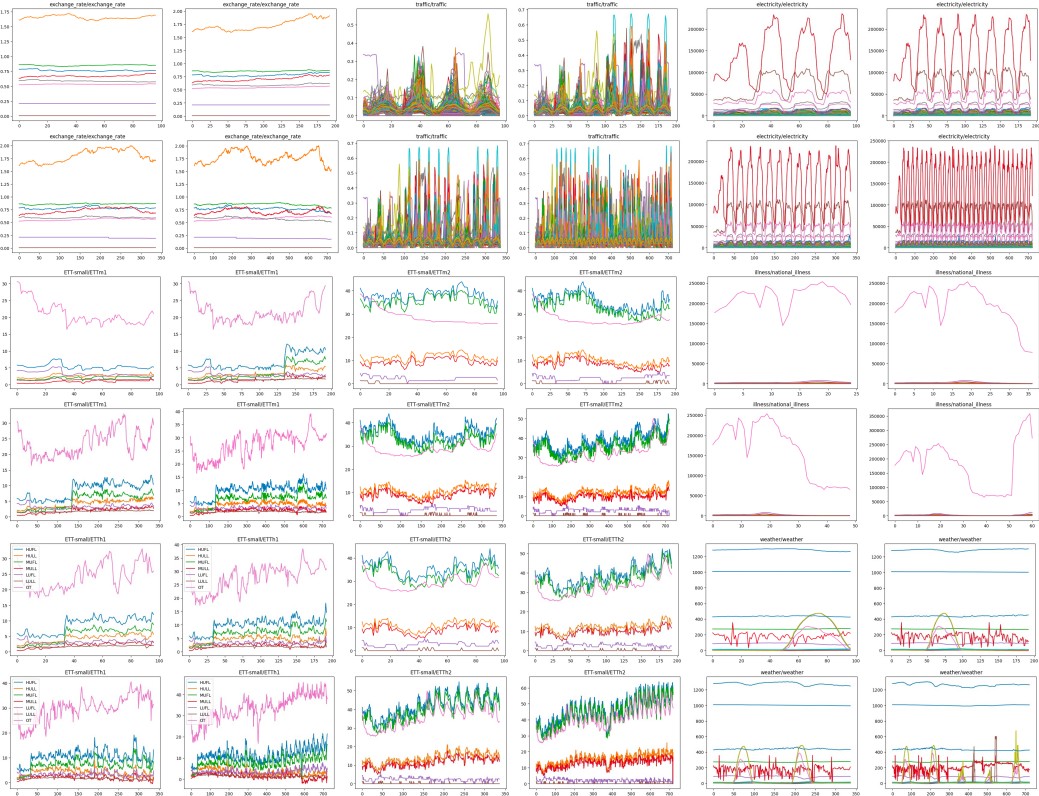

Figure 3: Time series samples extracted from the long-term forecasting dataset. The x-axis spans the pre-defined prediction window lengths within each dataset, with prediction lengths set to $T \in \{24, 36, 48, 60\}$ for the ILI dataset and $T \in \{96, 192, 336, 720\}$ for the remaining datasets.

**Normalized Mean Absolute Error (NMAE)** The Normalized Mean Absolute Error (NMAE) is a normalized version of the MAE, which is dimensionless and facilitates the comparability of the error magnitude across different datasets or scales. The mathematical representation of NMAE is given by:

$$\text{NMAE} = \frac{1}{K \times T} \sum_{i=1}^{K} \sum_{t=1}^{T} \frac{|x_{i,t} - \hat{x}_{i,t}|}{|x_{i,t}|}.$$

Its aggregated version is:

$$\text{NMAE}_{\text{sum}} = \frac{1}{T} \sum_{t=1}^{T} \frac{|x_t^{\text{sum}} - \hat{x}_t^{\text{sum}}|}{|x_t^{\text{sum}}|}.$$

**Mean Squared Error (MSE)** The Mean Squared Error (MSE) is a quantitative metric used to measure the average squared difference between the observed actual value and forecasts. It is defined mathematically as follows:

$$\text{MSE} = \frac{1}{K \times T} \sum_{i=1}^{K} \sum_{t=1}^{L} (x_{i,t} - \hat{x}_{i,t})^2.$$

For multivariate time series, we also provide the aggregated version:

$$\text{MSE}_{\text{sum}} = \frac{1}{T} \sum_{t=1}^{L} (x_t^{\text{sum}} - \hat{x}_t^{\text{sum}})^2.$$

**Normalized Root Mean Squared Error (NRMSE)**   The Normalized Root Mean Squared Error (NRMSE) is a normalized version of the Root Mean Squared Error (RMSE), which quantifies the average squared magnitude of the error between forecasts and observations, normalized by the expectation of the observed values. It can be formally written as:

$$\text{NRMSE} = \frac{\sqrt{\frac{1}{K \times T} \sum_{i=1}^{K} \sum_{t=1}^{L} (x_{i,t} - \hat{x}_{i,t})^2}}{\frac{1}{K \times T} \sum_{i=1}^{K} \sum_{t=1}^{T} |x_{i,t}|}.$$

For multivariate time series, we also provide the aggregated version:

$$\text{NRMSE}_{\text{sum}} = \frac{\sqrt{\frac{1}{T} \sum_{t=1}^{L} (x_t^{\text{sum}} - \hat{x}_t^{\text{sum}})^2}}{\frac{1}{T} \sum_{t=1}^{T} |x_t^{\text{sum}}|}.$$

### C.2 Distribution-level Metrics

**Continuous Ranked Probability Score (CRPS)**   The Continuous Ranked Probability Score (CRPS) (Matheson & Winkler, 1976) quantifies the agreement between a cumulative distribution function (CDF) $F$ and an observation $x$, represented as:

$$\text{CRPS} = \int_{\mathbb{R}} (F(z) - \mathbb{I}\{x \leq z\})^2 dz,$$

where $\mathbb{I}x \leq z$ denotes the indicator function, equating to one if $x \leq z$ and zero otherwise.

Being a proper scoring function, CRPS reaches its minimum when the predictive distribution $F$ coincides with the data distribution. When using the empirical CDF of $F$, denoted as $\hat{F}(z) = \frac{1}{n} \sum_{i=1}^{n} \mathbb{I}\{X_i \leq z\}$, where $n$ represents the number of samples $X_i \sim F$, CRPS can be precisely calculated from the simulated samples of the conditional distribution $p_\theta(\boldsymbol{x}_t | \boldsymbol{h}_t)$. In our practice, 100 samples are employed to estimate the empirical CDF.

For multivariate time series, the aggregate CRPS, denoted as $\text{CRPS}_{\text{sum}}$, is derived by summing across the $K$ time series, both for the ground-truth data and sampled data, and subsequently averaging over the forecasting horizon. Formally, it is represented as:

$$\text{CRPS}_{\text{sum}} = \mathbb{E}_t \left[ \text{CRPS} \left( \hat{F}_{\text{sum}}(t), \sum_{i=1}^{K} x_{i,l}^0 \right) \right].$$

## D Implementation Details

### D.1 Experiment settings

`ProbTS` is implemented using PyTorch Lightning (Falcon & The PyTorch Lightning team, 2019). During the training, we sample 100 batches per epoch and train for a maximum of 50 epochs, employing the CRPS metric as the monitor for checkpoint saving. We employ the Adam optimizer for all experiments, which are executed on single NVIDIA Tesla V100 GPUs using CUDA 11.3. In the evaluation phase, we sample 100 times to report the metrics on the test set.

### D.2 Hyper-parameters

We carried out an extensive grid search for models, tuning hyperparameters individually for each method. Given the large number of models, we inlude only the partial hyperparameter settings in Table 8. All hyperparameter configurations identified for each model on every dataset will be accessible via a GitHub repository, to be open-sourced subsequent to the paper's publication.

## E Additional Results and Experiments

### E.1 Impact of Data Scale

To further explore critical characteristics of time-series forecasting, we have examined the correlation between model performance gains, relative to the baseline model (GRU), and dataset dimensions,

Table 8: Hyperparameter settings for Electricity-S dataset.

| Model | Hyperparameter |
|---|---|
| DLinear | learning_rate=0.01, kernel_size=3, f_hidden_size=40 |
| PatchTST | learning_rate=0.0001, stride=3, patch_len=6, n_layers=3, n_heads=8, dropout=0.1, kernel_size=3, f_hidden_size=32 |
| TimesNet | learning_rate=0.001, n_layers=2, num_kernels=6, top_k=5, f_hidden_size=64, d_ff=64 |
| GRU NVP | learning_rate=0.001, f_hidden_size=40, num_layers=2, n_blocks=3, hidden_size=100, conditional_length=200 |
| GRU MAF | learning_rate=0.001, f_hidden_size=40, num_layers=2, n_blocks=4, hidden_size=100, conditional_length=200 |
| Trans MAF | learning_rate=0.001, f_hidden_size=32, num_heads=8, n_blocks=4, hidden_size=100, conditional_length=200 |
| TimeGrad | learning_rate=0.001, f_hidden_size=128, num_layers=4, conditional_length=100, beta_end=0.1, diff_steps=100 |
| CSDI | learning_rate=0.001, channels=64, emb_time_dim=128, emb_feature_dim=16, num_steps=50, num_heads=8, n_layers=4 |

length, and volume (see Table 9). However, our analysis does not identify a significant correlation between these factors and model performance.

Table 9: The correlation coefficient between the data volume and the relative performance improvement compared to the baseline model (GRU).

| Model | DLinear | | PatchTST | | GRU NVP | | TimeGrad | | CSDI | |
|---|---|---|---|---|---|---|---|---|---|---|
| | CRPS | NMAE | CRPS | NMAE | CRPS | NMAE | CRPS | NMAE | CRPS | NMAE |
| # Var. | 0.2422 | 0.2422 | -0.2676 | -0.2676 | -0.1856 | -0.2136 | -0.1665 | -0.1793 | -0.2315 | -0.2592 |
| # Total timestep | -0.1422 | -0.1422 | 0.3821 | 0.3821 | 0.3072 | 0.3329 | 0.2860 | 0.2971 | 0.3542 | 0.3826 |
| # Var. × Timestep | 0.0162 | 0.0162 | 0.0166 | 0.0166 | -0.0068 | -0.0011 | 0.0082 | 0.0117 | -0.0053 | -0.0133 |

## E.2 STATISTICAL AND GRADIENT BOOSTING DECISION TREE BASELINES

To enhance the empirical robustness of our study, we integrate classical statistical models, including ARIMA (Makridakis & Hibon, 1997) and ETS (Hyndman & Athanasopoulos, 2018), along with the Gradient Boosting Decision Tree (GBDT) model, XGBoost, into the ProbTS framework. The results in Table 10 clearly demonstrate the superior performance of deep learning methods over simple statistical baselines, emphasizing the importance of capturing non-linear dependencies for accurate forecasts. Notably, ARIMA and ETS exhibit varied performance across different data characteristics. ARIMA struggles with datasets like Solar, characterized by weak trending and strong seasonality, while ETS shows better adaptability. Conversely, in cases of strong trending and weak seasonality, as observed in the 'Wikipedia' dataset, ARIMA significantly outperforms ETS.

Utilizing the implementation from Elsayed et al. (2021), we find that XGBoost competes well, even surpassing neural network models in certain scenarios. However, for datasets with more complex distributions like 'Solar' and 'Electricity,' advanced probabilistic estimation methods demonstrate a substantial advantage over traditional learning methods and point estimation techniques. This highlights the adaptability and strength of advanced probabilistic methods in handling intricate forecasting scenarios.

## E.3 EXPERIMENTS ON UNIVARIATE DATASETS

In pursuit of a comprehensive analysis spanning univariate and multivariate scenarios, we examined a subset of M4 (Makridakis et al., 2020), M5 (Makridakis et al., 2022), and TOURISM datasets (Athanasopoulos et al., 2011)—crucial datasets for univariate time-series forecasting. Table 11 provides a quantitative assessment of the intrinsic characteristics of these new datasets, focusing on trending strength, seasonality, and data distribution complexity, as detailed in our paper. Notably, these datasets, except for M4-Daily may exhibit fewer seasonal patterns, do not introduce particularly unique characteristics.

Table 12 presents experimental results for representative methods, consistent with our initial observations. Probabilistic estimation methods like GRU NVP and TimeGrad excel on datasets with complex distributions (e.g., M4-Weekly and M5), while simpler point forecasting methods such

Table 10: Results of statistical models and GBDT baseline on short-term forecasting datasets.

| Model | Exchange Rate | | Solar | | Electricity | | Traffic | | Wikipedia | |
|---|---|---|---|---|---|---|---|---|---|---|
| | CRPS | NMAE | CRPS | NMAE | CRPS | NMAE | CRPS | NMAE | CRPS | NMAE |
| ARIMA | 0.009 | 0.009 | 1.000 | 1.000 | 0.164 | 0.164 | 0.461 | 0.461 | 0.348 | 0.348 |
| ETS | 0.011 | 0.011 | 0.580 | 0.580 | 0.121 | 0.121 | 0.413 | 0.413 | 0.685 | 0.685 |
| ETS-prob | 0.008 | 0.011 | 0.795 | 0.695 | 0.123 | 0.129 | 0.380 | 0.433 | 0.625 | 0.697 |
| XGBoost | 0.011 | 0.011 | 0.599 | 0.599 | 0.074 | 0.074 | 0.196 | 0.196 | - | - |
| DLinear | $0.012_{.001}$ | $0.012_{.001}$ | $0.547_{.009}$ | $0.547_{.009}$ | $0.095_{.006}$ | $0.095_{.006}$ | $0.273_{.012}$ | $0.273_{.012}$ | $1.046_{.037}$ | $1.046_{.037}$ |
| PatchTST | $\underline{0.010}_{.000}$ | $\mathbf{0.010}_{.000}$ | $0.496_{.002}$ | $0.496_{.002}$ | $0.076_{.001}$ | $0.076_{.001}$ | $0.202_{.001}$ | $0.202_{.001}$ | $\underline{0.257}_{.001}$ | $\mathbf{0.257}_{.001}$ |
| TimesNet | $0.011_{.001}$ | $0.011_{.001}$ | $0.507_{.019}$ | $0.507_{.019}$ | $0.071_{.002}$ | $0.071_{.002}$ | $0.205_{.002}$ | $0.205_{.002}$ | $0.304_{.002}$ | $0.304_{.002}$ |
| GRU NVP | $0.016_{.003}$ | $0.020_{.003}$ | $0.396_{.021}$ | $0.507_{.022}$ | $0.055_{.002}$ | $0.073_{.003}$ | $0.161_{.006}$ | $0.203_{.009}$ | $0.282_{.003}$ | $0.330_{.003}$ |
| GRU MAF | $0.015_{.001}$ | $0.020_{.001}$ | $0.386_{.026}$ | $0.492_{.027}$ | $\underline{0.051}_{.001}$ | $\underline{0.067}_{.001}$ | $\underline{0.131}_{.006}$ | $0.165_{.009}$ | $0.281_{.004}$ | $0.337_{.005}$ |
| Trans MAF | $0.011_{.001}$ | $0.014_{.001}$ | $0.400_{.022}$ | $0.503_{.022}$ | $0.054_{.004}$ | $0.071_{.005}$ | $\mathbf{0.129}_{.004}$ | $\mathbf{0.165}_{.006}$ | $0.289_{.008}$ | $0.344_{.008}$ |
| TimeGrad | $0.011_{.001}$ | $0.014_{.002}$ | $\mathbf{0.359}_{.011}$ | $\mathbf{0.445}_{.023}$ | $0.052_{.001}$ | $0.067_{.001}$ | $0.164_{.091}$ | $0.201_{.115}$ | $0.272_{.008}$ | $0.327_{.011}$ |
| CSDI | $\mathbf{0.008}_{.000}$ | $0.011_{.000}$ | $0.366_{.005}$ | $0.484_{.008}$ | $\mathbf{0.050}_{.001}$ | $\mathbf{0.065}_{.001}$ | $0.146_{.012}$ | $0.176_{.013}$ | $\mathbf{0.219}_{.006}$ | $0.259_{.009}$ |

Table 11: Quantitative assessment of the intrinsic characteristics of the univariate datasets. The JS Div denotes Jensen–Shannon divergence, where a lower score indicates closer approximations to a Gaussian distribution.

| Dataset | M4-Weekly | M4-Daily | M5 | TOURISM-Monthly |
|---|---|---|---|---|
| Trend $F_T$ | 0.7677 | 0.9808 | 0.3443 | 0.7979 |
| Seasonality $F_S$ | 0.3401 | 0.0467 | 0.2480 | 0.6826 |
| JS Div. | 0.5106 | 0.4916 | 0.6011 | 0.3291 |

as DLinear and PatchTST perform well on datasets with relatively simple data distribution, like TOURISM-Monthly. Both autoregressive and non-autoregressive decoding schemes show comparable performance in short-term forecasting, as discussed in the main paper."

Table 12: Results on M4, M5, and TOURISM datasets. We utilize a lookback window of 3H, with 'H' denoting the forecasting horizon.

| Model | DLinear | | PatchTST | | GRU NVP | | TimeGrad | |
|---|---|---|---|---|---|---|---|---|
| | CRPS | NMAE | CRPS | NMAE | CRPS | NMAE | CRPS | NMAE |
| M4-Weekly | 0.081 | 0.081 | 0.089 | 0.089 | 0.066 | 0.077 | 0.055 | 0.065 |
| M4-Daily | 0.034 | 0.034 | 0.035 | 0.035 | 0.030 | 0.038 | 0.026 | 0.032 |
| M5 | 0.891 | 0.891 | 0.898 | 0.898 | 0.679 | 0.864 | - | - |
| TOURISM-Monthly | 0.168 | 0.168 | 0.136 | 0.136 | 0.171 | 0.223 | 0.152 | 0.191 |

### E.4 EXPERIMENTS ON SYNTHETIC DATASETS

To enhance the rigor of the insights presented, we employ synthetic datasets, encompassing a baseline dataset and variants with pronounced trends, strong seasonality, and complex data distribution (see Table 13). Each dataset comprises series generated by combining trend, seasonality, noise, and anomaly components with controlled characteristics. Subsequent experiments on these synthetic datasets (refer to Table 14), using representative models, validate the empirical findings established on other datasets with ProbTS. Key observations include the declining performance of autoregressive decoding models, such as TimeGrad, in the presence of increasing trends, improved performance for models using autoregressive decoding with intensifying seasonality, and the competitive performance of probabilistic methods like CSDI in handling more complex data distributions.

### E.5 CASE STUDY

To intuitively demonstrate the distinct characteristics of point and probabilistic estimations, a case study was conducted on short-term datasets. Figure 4 illustrates that point estimation yields single-valued, deterministic estimates, in contrast to probabilistic methods, which model continuous data distributions as depicted in Figure 5. This modeling of data distributions captures the uncertainty

Table 13: Quantitative assessment of intrinsic characteristics for synthetic datasets. The JS Div denotes Jensen–Shannon divergence, where a lower score indicates closer approximations to a Gaussian distribution.

| Dataset | Normal | Strong Trend | Strong Seasonality | Complex Distribution |
|---|---|---|---|---|
| Trend $F_T$ | 0.105 | 0.554 | 0.105 | 0.064 |
| Seasonality $F_S$ | 0.302 | 0.302 | 0.791 | 0.190 |
| JS Div. | 0.261 | 0.248 | 0.272 | 0.469 |

Table 14: Results on synthetic datasets. The look-back window and forecasting horizon are 30.

| Model | Normal | | Strong Trend | | Strong Seasonality | | Complex Distribution | |
|---|---|---|---|---|---|---|---|---|
| | CRPS | NMAE | CRPS | NMAE | CRPS | NMAE | CRPS | NMAE |
| DLinear | 0.013 | 0.013 | **0.001** | **0.001** | 0.014 | 0.014 | 0.301 | 0.301 |
| PatchTST | **0.012** | **0.012** | **0.001** | **0.001** | **0.012** | **0.012** | 0.275 | **0.275** |
| TimeGrad | 0.024 | 0.032 | 0.042 | 0.048 | 0.022 | 0.028 | 0.283 | 0.338 |
| CSDI | 0.013 | 0.014 | 0.010 | 0.007 | 0.020 | 0.027 | **0.269** | 0.301 |

in forecasts, aiding decision-makers in fields such as weather and finance to make more informed choices. It is also observed that while both methods align well with ground truth values in short-term forecasting datasets, they struggle to accurately capture outliers, particularly noted in the Wikipedia dataset.

### E.6 MODEL EFFICIENCY

For reference, detailed results regarding memory usage and time efficiency for five representative models on long-term forecasting datasets are provided here. Table 15 displays the computation memory of various models with a forecasting horizon set to 96. Additionally, Table 16 compares the inference time of these models on long-term forecasting datasets, illustrating the impact of changes in the forecasting horizon.

Table 15: Computation memory. The batch size is 1 and the prediction horizon is set to 96.

| Metric | Dataset | DLinear | PatchTST | LSTM NVP | TimeGrad | CSDI |
|---|---|---|---|---|---|---|
| NPARAMS (MB) | ETTm1 | 0.075 | 2.145 | 1.079 | 1.233 | 1.720 |
| | Electricity | 0.076 | 2.146 | 3.680 | 3.472 | 1.370 |
| | Traffic | 0.078 | 2.149 | 15.926 | 8.298 | 1.390 |
| | Weather | 0.075 | 2.145 | 3.085 | 0.574 | 1.721 |
| | Exchange | 0.075 | 0.135 | 1.979 | 0.488 | 1.720 |
| Max GPU Mem. (GB) | ETTm1 | 0.002 | 0.009 | 0.010 | 0.012 | 0.027 |
| | Electricity | 0.060 | 0.068 | 0.129 | 0.128 | 1.411 |
| | Traffic | 0.161 | 0.168 | 0.361 | 0.333 | 9.102 |
| | Weather | 0.004 | 0.012 | 0.021 | 0.012 | 0.070 |
| | Exchange | 0.002 | 0.002 | 0.013 | 0.008 | 0.030 |

## F FURTHER DISCUSSION ON CROSS-CHANNEL INTERACTIONS

We compile a summary table (Table 17) delineating how models from each branch address the multivariate aspect. Despite a thorough investigation, we have not identified a clear pattern linking the modeling of cross-channel interactions to overall model performance. A notable trend is the prevalent use of a channel-mixing approach in most studies. However, findings are diverse; models like DLinear and PatchTST suggest that processing channels independently can yield superior results, while others like CSDI indicate that explicit modeling of cross-channel interactions offers signifi-

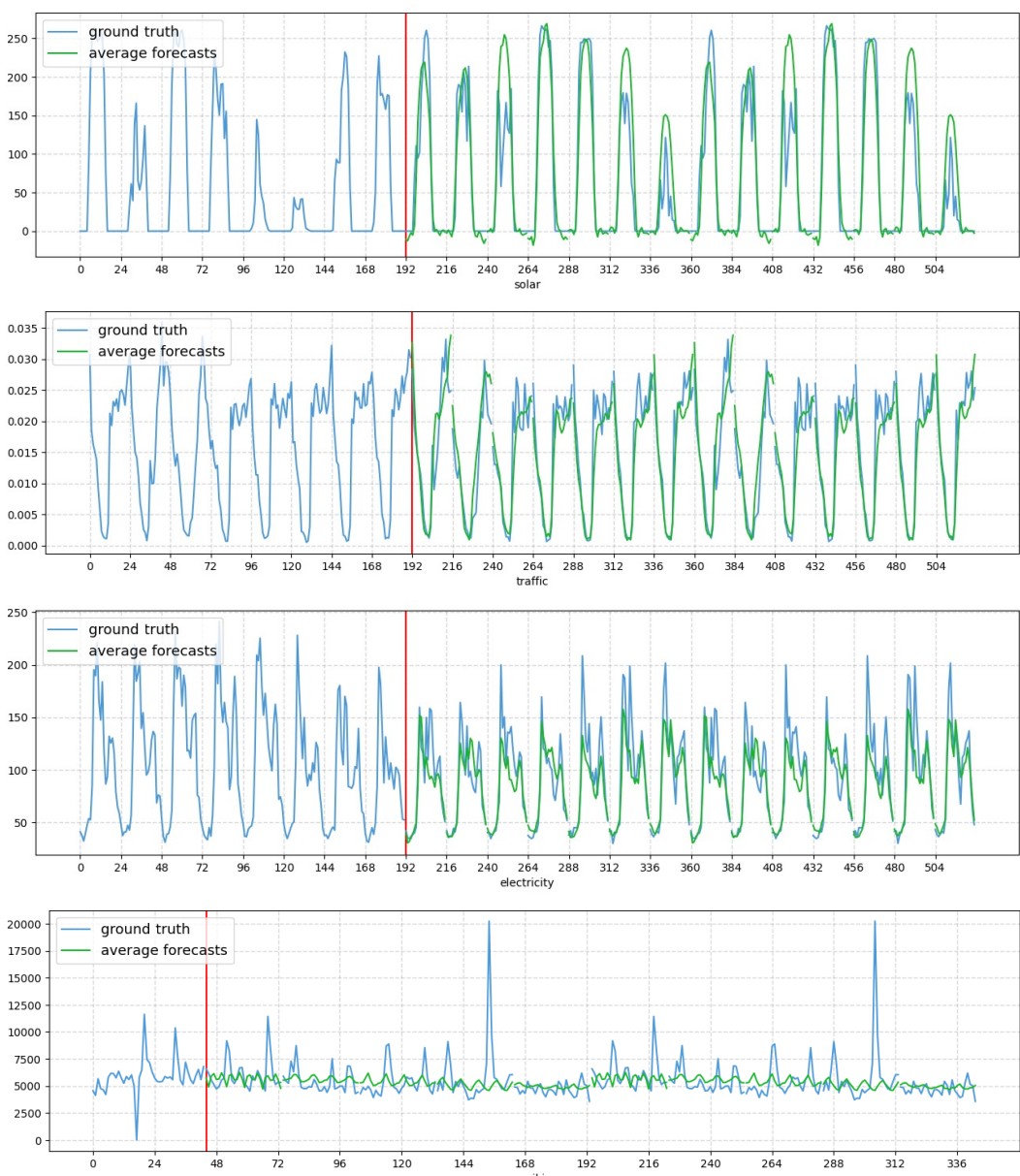

Figure 4: Point forecasts from the PatchTST model and the ground-truth value on short-term forecasting datasets.

cant advantages. This diversity underscores the ongoing exploration of the impact of cross-channel interactions on forecasting performance.

Figure 5: Forecasting intervals from the TimeGrad model and the ground-truth value on short-term forecasting datasets.

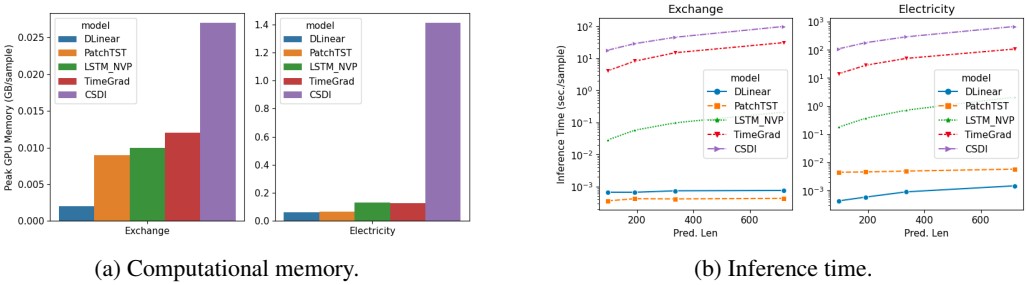

(a) Computational memory.

(b) Inference time.

Figure 6: Comparison of computational efficiency. The forecasting horizon is set to 96 for calculating memory usage.

Table 16: Comparison of inference time (sec./sample).

| Model | pred len | DLinear | PatchTST | LSTM NVP | TimeGrad | CSDI |
|---|---|---|---|---|---|---|
| ETTm1 | 96 | $0.0003 \pm 0.0000$ | $0.0003 \pm 0.0000$ | $0.0352 \pm 0.0007$ | $4.1067 \pm 0.0504$ | $16.3280 \pm 0.0747$ |
| | 192 | $0.0003 \pm 0.0000$ | $0.0003 \pm 0.0000$ | $0.0697 \pm 0.0020$ | $7.8979 \pm 0.0403$ | $25.8378 \pm 0.3124$ |
| | 336 | $0.0003 \pm 0.0000$ | $0.0003 \pm 0.0000$ | $0.1221 \pm 0.0044$ | $13.6197 \pm 0.1023$ | $39.8832 \pm 0.2157$ |
| | 720 | $0.0004 \pm 0.0000$ | $0.0003 \pm 0.0000$ | $0.2603 \pm 0.0020$ | $28.6074 \pm 1.1346$ | $86.1862 \pm 0.1863$ |
| Electricity | 96 | $0.0004 \pm 0.0000$ | $0.0045 \pm 0.0001$ | $0.1783 \pm 0.0006$ | $13.8439 \pm 0.0054$ | $388.3150 \pm 0.2155$ |
| | 192 | $0.0006 \pm 0.0000$ | $0.0046 \pm 0.0000$ | $0.3700 \pm 0.0010$ | $27.6683 \pm 0.0368$ | $659.4284 \pm 0.2003$ |
| | 336 | $0.0008 \pm 0.0000$ | $0.0049 \pm 0.0000$ | $0.7157 \pm 0.0028$ | $48.4456 \pm 0.0279$ | - |
| | 720 | $0.0015 \pm 0.0000$ | $0.0057 \pm 0.0000$ | $2.0785 \pm 0.0186$ | $104.1473 \pm 0.1465$ | - |
| Traffic | 96 | $0.0010 \pm 0.0001$ | $0.0102 \pm 0.0000$ | $0.3695 \pm 0.0022$ | $31.7644 \pm 0.0101$ | - |
| | 192 | $0.0013 \pm 0.0000$ | $0.0106 \pm 0.0000$ | $0.8287 \pm 0.0094$ | $63.5832 \pm 0.0060$ | - |
| | 336 | $0.0020 \pm 0.0000$ | $0.0114 \pm 0.0001$ | $1.6945 \pm 0.0026$ | $111.4147 \pm 0.0169$ | - |
| | 720 | $0.0039 \pm 0.0000$ | $0.0137 \pm 0.0000$ | $5.0963 \pm 0.0018$ | $258.1274 \pm 0.6088$ | - |
| Weather | 96 | $0.0002 \pm 0.0000$ | $0.0004 \pm 0.0000$ | $0.0800 \pm 0.0016$ | $4.1261 \pm 0.0812$ | $37.8984 \pm 0.0782$ |
| | 192 | $0.0003 \pm 0.0000$ | $0.0004 \pm 0.0000$ | $0.1568 \pm 0.0008$ | $8.2913 \pm 0.5544$ | $62.0223 \pm 0.2329$ |
| | 336 | $0.0003 \pm 0.0000$ | $0.0004 \pm 0.0000$ | $0.2482 \pm 0.0297$ | $14.2391 \pm 0.4891$ | $96.8704 \pm 0.2258$ |
| | 720 | $0.0003 \pm 0.0000$ | $0.0005 \pm 0.0000$ | $0.5447 \pm 0.0249$ | $29.4407 \pm 0.3519$ | $216.6044 \pm 0.4253$ |
| Exchange | 96 | $0.0006 \pm 0.0000$ | $0.0004 \pm 0.0000$ | $0.0284 \pm 0.0001$ | $4.1069 \pm 0.0981$ | $17.8655 \pm 0.1282$ |
| | 192 | $0.0007 \pm 0.0000$ | $0.0004 \pm 0.0000$ | $0.0563 \pm 0.0008$ | $8.1576 \pm 0.0911$ | $28.5456 \pm 0.0873$ |
| | 336 | $0.0007 \pm 0.0000$ | $0.0004 \pm 0.0000$ | $0.0966 \pm 0.0007$ | $14.4593 \pm 0.4466$ | $44.9733 \pm 0.3820$ |
| | 720 | $0.0007 \pm 0.0000$ | $0.0004 \pm 0.0000$ | $0.2085 \pm 0.0046$ | $30.1443 \pm 0.5378$ | $97.7417 \pm 0.2606$ |
| ILI | 24 | $0.0002 \pm 0.0000$ | $0.0008 \pm 0.0001$ | $0.0080 \pm 0.0001$ | $1.0427 \pm 0.0190$ | $12.4038 \pm 0.1681$ |
| | 192 | $0.0002 \pm 0.0000$ | $0.0008 \pm 0.0000$ | $0.0121 \pm 0.0003$ | $1.5762 \pm 0.0282$ | $12.7187 \pm 0.1344$ |
| | 336 | $0.0002 \pm 0.0000$ | $0.0008 \pm 0.0000$ | $0.0155 \pm 0.0002$ | $2.1344 \pm 0.0660$ | $12.7386 \pm 0.1868$ |
| | 720 | $0.0002 \pm 0.0000$ | $0.0008 \pm 0.0000$ | $0.0196 \pm 0.0004$ | $2.5787 \pm 0.0594$ | $12.5407 \pm 0.0481$ |

Table 17: Summary of how existing models handle multivariate time series.

| Model | Research branch | Process channels independently |
|---|---|---|
| Customized neural architectures | N-BEATS (Oreshkin et al., 2020) | ✓ |
| | N-HiTS (Challu et al., 2023) | ✓ |
| | Autoformer (Wu et al., 2021) | ✗ |
| | Informer (Zhou et al., 2021) | ✗ |
| | LTSF-Linear (Zeng et al., 2023) | ✗/✓ |
| | PatchTST (Nie et al., 2023) | ✗/✓ |
| | TimesNet (Wu et al., 2023) | ✗ |
| Probabilistic estimation | DeepAR (Salinas et al., 2020) | ✓ |
| | GP-copula (Salinas et al., 2019) | ✗ |
| | LSTM NVP (Rasul et al., 2021b) | ✗ |
| | LSTM MAF (Rasul et al., 2021b) | ✗ |
| | Trans MAF (Rasul et al., 2021b) | ✗ |
| | TimeGrad (Rasul et al., 2021a) | ✗ |
| | CSDI (Tashiro et al., 2021) | ✗ |
| | SPD (Bilos et al., 2023) | ✗ |

