# OpenReview forum: "ProbTS: A Unified Toolkit to Probe Deep Time-series Forecasting"
_ICLR.cc/2024/Conference — Submitted to ICLR 2024_

### Official Review · Reviewer_kFUJ · 2023-10-30

**Soundness:** 2 fair
**Presentation:** 1 poor
**Contribution:** 2 fair
**Rating:** 3
**Confidence:** 5

**Summary:**

The paper highlights two main directions of deep learning for time series forecasting - architecture design, and probabilistic forecasting heads. They present a new library which attempts to address both directions, and present some benchmark results and empirical studies.

**Strengths:**

The paper presents a nice position and overview on the research directions for time series forecasting within the deep learning community.

**Weaknesses:**

Unfortunately, this paper tries to do too much and too little at the same time.
1. As a paper introducing a new library, there is insufficient details of the design and implementation of the library. It also has insufficient comparison with existing libraries - what sets it apart from existing work?

    a. Table 1 does not really make sense -- the header for column 1 is "Model", all the comparisons are different models, but ProbTS is not a model. It would make more sense to compare ProbTS with other libraries/packages (e.g. GluonTS, TSLib, etc.) rather than specific models/papers.

    b. More attributes for libraries should be compared -- metrics, datasets, data transformations, data loading efficiency, ...

    c. More library comparisons should be added [1, 2, 3], and many others.

    d. The characterization of GluonTS as "each specializing in a single forecasting paradigm, fall short of our research objective to unify the two distinct research branches" is not accurate -- new architectures can and have been implemented in it. Also see how it has been used in [4].

2. As a benchmark paper, it fails to perform a comprehensive evaluation in both dimensions of architecture design and probabilistic forecasting head.

    a. In Table 4, only a small number of methods from each dimension has been evaluated on.

    b. A more comprehensive evaluation, combining different architectures with different probabilistic forecasting heads can be presented.

3. As a an empirical study, it does not yield any definitive insights into the interplay between architecture design and probabilistic forecasting head.

    a. More insights regarding various architecture designs should be given -- e.g. for architectures like Autoformer -- how can be attach probabilistic heads, since the architecture design outputs the prediction based on seasonality + trend? What about PatchTST, how does patching affect probabilistic heads?

Note that I am not saying the paper should achieve everything mentioned above, but one particular direction should be chosen to go all in.

[1] https://github.com/unit8co/darts

[2] https://github.com/salesforce/Merlion

[3] https://github.com/facebookresearch/Kats

[4] https://github.com/kashif/pytorch-transformer-ts

**Questions:**

None

---

> ### Author Response · Authors · 2023-11-20
> **Response to Reviewer kFUJ (Part 1/2)**
>
> Thank you for your thorough review of our paper. We apologize for any misunderstanding and lack of clarity regarding the strengths of our work. We hope the following clarifications and explanations can help address your concerns.
>
> **Response to Weakness 1**
>
> We would like to clarify that the primary contribution of this paper is to unify two key aspects in deep time-series forecasting: customized neural architecture designs and advanced probabilistic estimations tailored for time series. **This work is set apart from existing studies in that we strive to unify these two aspects and expose underexplored yet crucial topics, accompanied by our findings and insights, to the research community**. Specifically, this paper discloses and analyzes several unanswered questions:
>
> - How do different data characteristics and forecasting horizons influence methodological designs in previous studies? What are the implications when evaluating them in underexplored scenarios?
> - What are the strengths and weaknesses of either focusing on neural architecture designs or emphasizing advanced probabilistic estimations? How will existing approaches perform on different evaluation metrics, when forecasting distributional or point-level forecasts?
> - Why does the development of these two research branches lead to different preferences on decoding schemes, either autoregressive or non-autoregressive?
>
> We've briefly summarized our insights towards these questions in the introduction of our paper and included more detailed results and analyses in Section 4. What we want to clarify and emphasize most is that **ProbTS is not a traditional tool** aiming to consolidate all library attributes, such as metrics, datasets, data transformations, and data loading efficiency, into one place. Instead, we envision **ProbTS as a unique research-oriented tool** that unifies divergent yet equally important research in deep time-series forecasting, reveals overlooked challenges, and opens up new avenues for future research.
>
> In response to your constructive suggestions regarding the wording of the "Model" column in Table 1 and the additional discussions between this work and existing tools, we have made appropriate adjustments. For instance, we have changed the "Model" column to "Studies", indicating that each entry in this column represents an individual paper studying specific problems. We have also compared ProbTS with existing tools from various perspectives, as shown in the following table.
>
> | Tools | Provide datasets | Short-term Benchmarking | Long-term Benchmarking | Prob. Forecasting | Advanced NN Arch. |
> | :--- | :---: | :---: | :---: | :---: | :---: |
> | Darts [4] | √ | x | x | √ | √ |
> | Merlion [5] | √ | √ | x | x | √ |
> | Kats [6] | x | x | x | x | x |
> | pytorch-transformer-ts [3] | √ | x | x | x | √ |
> | TSlib [1] | √ | √ | √ | x | √ |
> | GluonTS [2] | √ | √ | x | √ | x |
> | ProbTS | √ | √ | √ | √ | √ |
>
> Indeed, GluonTS is an excellent tool that comes closest to ours. However, when we attempt to unify two different research branches, we find that GluonTS is highly wrapped and leans toward probabilistic forecasting, making it difficult to incorporate different data preprocessing, normalization, and evaluation implementations used by existing studies in neural architecture designs. As a result, we decided to develop ProbTS from scratch as a research-oriented tool specializing in unifying neural architecture designs and probabilistic estimations in deep time-series forecasting.
>
> [1] Alexandrov, Alexander, et al. "Gluonts: Probabilistic time series models in python." *arXiv preprint arXiv:1906.05264* (2019).
>
> [2] [https://github.com/thuml/Time-Series-Library](https://github.com/thuml/Time-Series-Library)
>
> [3] [https://github.com/kashif/pytorch-transformer-ts](https://github.com/kashif/pytorch-transformer-ts)
>
> [4] [https://github.com/unit8co/darts](https://github.com/unit8co/darts)
>
> [5] [https://github.com/salesforce/Merlion](https://github.com/salesforce/Merlion)
>
> [6] [https://github.com/facebookresearch/Kats](https://github.com/facebookresearch/Kats)

---

> ### Author Response · Authors · 2023-11-20
> **Response to Reviewer kFUJ (Part 2/2)**
>
> **Response to weakness 2 & 3**
>
> Thank you for your comments. We appreciate your observations and would like to emphasize that our evaluation does indeed provide a robust comparison of existing methods across two distinct research branches.
>
> Firstly, for our benchmarking, we have included the most representative and top-performing baselines from each branch. This includes highly competitive neural architectures such as PatchTST, TimesNet, D-linear, N-HiTS, as well as recent advanced probabilistic forecasting approaches like CSDI, TimeGrad, and Trans MAF. We are certainly open to incorporating additional methods into our comparisons. If you could suggest any that surpass the performance or possess unique advantages over these existing baselines, we would be more than willing to consider them.
>
> Secondly, we've created some ablation variants by removing the probabilistic head from GRU NVP and Trans MAF, which are represented by the GRU and Transformer models in Table 4. Concerning your proposal of stacking probabilistic heads on customized architectures, such as Autoformer and PatchTST, it's crucial to note that this isn't a simple undertaking. It either necessitates transforming these non-autoregressive architectures to the autoregressive scheme (as seen in GRU MAF, GRU NVP, Trans MAF, TimeGrad) or integrating probabilistic modules in a non-autoregressive manner. For example, CSDI combines architecture designs with non-autoregressive probabilistic estimations. However, CSDI struggles in long-term forecasting due to its increasing inefficiency as the forecasting horizon expands, and its inability to handle long trending or strong seasonality effectively.
>
> In summary, the task of combining customized architectures with advanced probabilistic estimations presents many unresolved challenges and is not as simple as adding a probabilistic head over an existing architecture. After disclosing and analyzing these remaining problems in this paper, we believe there are many opportunities ahead to explore how we can better combine customized architectures with advanced probabilistic estimations while addressing the limitations of existing probabilistic forecasting methods. In this regard, ProbTS aims to serve as a foundational tool to facilitate future research.

---

> ### Comment · Reviewer_kFUJ · 2023-11-21
>
> Thank you for the response. My thoughts and score currently remains the same.
>
> 1. Could you explain concretely what you mean by "ProbTS as a unique research-oriented tool that unifies divergent yet equally important research in deep time-series forecasting, reveals overlooked challenges, and opens up new avenues for future research."? What are the features or toolkit design that allows ProbTS to be so unique?
> 2. I disagree with the characterization that "we find that GluonTS is highly wrapped and leans toward probabilistic forecasting, making it difficult to incorporate different data preprocessing, normalization, and evaluation implementations used by existing studies in neural architecture design". GluonTS has an extensive data pre-processing pipeline (https://github.com/awslabs/gluonts/tree/dev/src/gluonts/transform), supports many normalization techniques (https://github.com/awslabs/gluonts/blob/dev/src/gluonts/torch/scaler.py), and has a full suite of evaluation metrics (https://github.com/awslabs/gluonts/tree/dev/src/gluonts/ev). It is also easily extendible, allowing to add new transforms, models, and evaluation metrics.
> 3. It is difficult to evaluate the benefits of ProbTS without looking at the code or any design documents about the library.
> 4. GluonTS already implements 2 of 4 of the "advanced neural architectures" mentioned, PatchTST and DLinear.
> 5. pytorch-transformer-ts uses GluonTS to implement many more "advanced neural architectures", and it does probabilistic forecasting.
> 6. The paper is very limited by not analyzing the interplay between architecture and probabilistic head. How can we disambiguate the effects of architecture and probabilistic head? Does performance stem from different architecture or head? Yes, it is not straightforward, but that would constitute a significant contribution rather than simply benchmarking different methods which arguably are not fair comparisons with each other.

---

> > ### Author Response · Authors · 2023-11-21
> > **Response to Official Comment by Reviewer kFUJ (Part 1/3)**
> >
> > Thank you for your valuable feedback. We acknowledge that our initial responses may not have fully addressed your concerns. We greatly appreciate your unique perspectives and understand that you are assessing this paper from a specific viewpoint. We aim to engage in a detailed discussion to highlight the key strengths and singular position of this paper.
> >
> > After a meticulous review of your additional queries, we have structured our responses into three segments.
> >
> > **Clarifying the Unique Contributions of this paper (In response to Question 1)**
> >
> > The primary objective of this work is to **bridge two significant yet distinct research branches within deep time-series forecasting**:
> >
> > - One branch focuses on customized neural architecture designs but limits itself to point-level forecasts.
> > - The other branch specializes in probabilistic estimations for distributional forecasts but rarely includes advanced architecture designs.
> >
> > Recognizing **this distinctive gap**, we developed ProbTS to cater to the specific requirements of such a unification study. Particularly, **we identified several crucial aspects that necessitate unification**, which include:
> >
> > - The role of different data characteristics (trend, seasonality, complexity of data distribution) and forecasting setups (long-term vs short-term) in influencing the methodological designs and choices.
> > - Crucial aspects in methological designs that may prefer different data scenarios.
> >     - Probabilistic vs non-probabilistic forecasting paradigms and evaluation metrics.
> >     - Customized vs general neural architecture designs.
> >     - Autoregressive vs non-autoregressive decoding schemes.
> >
> > Utilizing ProbTS, we have **unearthed unique insights and highlighted previously unnoticed challenges** in these areas:
> >
> > - Regarding data scenarios, we find that long-term forecasting scenarios predominantly exhibit more significant trending and seasonality than short-term forecasting scenarios, while
> > the latter typically display more complicated data distributions. These differences in data
> > characteristics greatly influence the method designs of those two research branches.
> > - We have identified the pros and cons of different methodological focuses. Probabilistic
> > forecasting methods indeed excel in modeling complex data distributions. However, we
> > find that they may produce poor point forecasts even with superior CRPS scores. Additionally, customized network architectures, specifically addressing the challenges induced
> > by trending and seasonality, perform remarkably well in long-term forecasting scenarios.
> > Nevertheless, effective network architectures for short-term forecasting scenarios remain
> > underexplored. These findings imply that there is a vast space to explore in future research,
> > especially in combining these two research branches across various data scenarios.
> > - Concerning different decoding schemes, we find that the autoregressive scheme excels in
> > cases with strong seasonality but struggles with pronounced trending, explaining why most
> > long-term forecasting studies consistently prefer the non-autoregressive scheme. Furthermore, we observe that both decoding schemes perform equally well in short-term forecasting, which also explains the diverse preferences for these two schemes in short-term forecasting studies. Looking into the future, we also expect new autoregressive decoding schemes to address the error-propagation challenge when facing significant trending.
> >
> > Given these explanations, we wish to underscore that **the primary strengths of ProbTS rest on its unique position as a unification study, the derived insights, and the uncovered opportunities for future research**. Consequently, it is not a particular toolkit design or feature that distinguishes ProbTS, but its singular position in research, its novel findings and insights, as well as the challenges and opportunities it uniquely uncovers.

---

> > ### Author Response · Authors · 2023-11-21
> > **Response to Official Comment by Reviewer kFUJ (Part 2/3)**
> >
> > **Rationale for Developing ProbTS to Achieve our Research Objectives, Rather than Directly Utilizing GluonTS or Other Tools (In response to Questions 2, 3, 4, 5)**
> >
> > Undoubtedly, GluonTS is a commendable framework for probabilistic time-series modeling in Python, boasting over 3.9k stars on Github. However, **GluonTS does not incorporate numerous advanced models featured in our study**. We have compiled a comparison of model coverage between GluonTS and ProbTS below, demonstrating that ProbTS includes a wider range of cutting-edge models. (We acknowledge that GluonTS includes many more traditional models, which are not listed here as our focus is on state-of-the-art methods.)
> >
> > | Model | GluonTS | ProbTS |
> > | --- | --- | --- |
> > | N-HiTS | x | √ |
> > | NLinear | x | √ |
> > | DLinear | √ | √ |
> > | PatchTST | √ | √ |
> > | TimesNet | x | √ |
> > | GRU NVP | x | √ |
> > | GRU MAF | x | √ |
> > | Trans MAF | x | √ |
> > | TimeGrad | x | √ |
> > | CSDI | x | √ |
> >
> > To fulfill the research objective of this paper, we required a research-oriented tool that not only implements the models from a wide array of previous studies but also meticulously aligns the experimental setups (such as data splits, pre-processing (e.g., global normalization), batch-level normalization (e.g., RevIN), and evaluation) with them to ensure we could reproduce their results. Specifically, we referred to several public repositories to ensure our setups were correctly aligned with those in the original papers. Here are some representative examples:
> >
> > - [https://github.com/zalandoresearch/pytorch-ts](https://github.com/zalandoresearch/pytorch-ts), built upon Gluonts, includes detailed experiments of various probabilistic methods, such as GRU NVP, GRU MAF, Trans MAF, and TimeGrad, on short-term forecasting scenarios.
> > - [https://github.com/ermongroup/CSDI](https://github.com/ermongroup/CSDI) includes the official implementation of CSDI.
> > - We predominantly referred to TSlib, [https://github.com/thuml/Time-Series-Library/](https://github.com/thuml/Time-Series-Library/blob/main/models/PatchTST.py), to implement relevant models, such as TimesNet, and to reproduce the experiments on long-term forecasting benchmarks.
> > - However, we noted that PatchTST ([https://github.com/thuml/Time-Series-Library/blob/main/models/PatchTST.py](https://github.com/thuml/Time-Series-Library/blob/main/models/PatchTST.py)) included by TSlib has certain issues, not using RevIN and channel-independence, which significantly influence performance. Therefore, we cross-referenced with the official implementation, [https://github.com/yuqinie98/PatchTST](https://github.com/yuqinie98/PatchTST), to ensure reproducibility.
> > - We also refer to the official implementation of N-HiTS ([https://github.com/cchallu/n-hits](https://github.com/cchallu/n-hits)) to ensure reproducibility.
> >
> > However, to the best of our knowledge, **GluonTS, being a general-purpose time-series package, does not include detailed processes or specifications to reproduce the paper-reported results of these models**. Similarly, pytorch-transformer-ts only partially meets these needs.
> >
> > Given the significant efforts required to align with two divergent branches of methods in terms of preprocessing, hyperparameter tuning, and evaluation aspects, we opted to maintain a self-contained codebase. We've also organized the code structure to reflect the unique comparisons this paper aims to highlight, such as long-term vs short-term scenarios, probabilistic vs non-probabilistic forecasts, customized vs standard architectures, and autoregressive vs non-autoregressive schemes.
> >
> > You can refer to the design of ProbTS in Figure 1. The code repository is currently undergoing an internal review process, and we assure it will be made publicly available upon completion.
> >
> > It's important to note that we do not present this tool as a comprehensive framework covering all aspects. Instead, we see it as an agile, research-oriented tool tailored to our specific research objectives. Therefore, **we kindly suggest you evaluate this work based on the research focus of this paper and our unique findings derived by utilizing this tool**.

---

> > ### Author Response · Authors · 2023-11-21
> > **Response to Official Comment by Reviewer kFUJ (Part 3/3)**
> >
> > **The Interplay between Architecture and Probabilistic Head (In response to Question 6)**
> >
> >
> > We are grateful for your insightful query regarding the interplay between architecture and the probabilistic head.
> >
> > Initially, we would like to draw your attention to Table 4, which offers some hints on this topic. In this table, GRU NVP, GRU MAF, and Trans MAF represent specific architectures incorporating a probabilistic head, while GRU and Transformer denote the base architecture where the probabilistic head is substituted with a linear projection. These examples illustrate the effectiveness of the probabilistic head.
> >
> > However, it's crucial to note that all these models adopt a straightforward design, which stacks a probabilistic module atop an encoding architecture. The design becomes significantly more complex in advanced non-autoregressive probabilistic methods such as CSDI, which integrate architecture design into a diffusion module. In such cases, there isn't an encoder, only a forecaster, thus eliminating the concept of a probabilistic head.
> >
> > Additionally, when it comes to non-autoregressive, customized neural architectures like PatchTST and TimesNet, we do not delve into how to transform them into effective probabilistic forecasters. To our knowledge, this is not a simple process and warrants comprehensive investigations and explorations of various possibilities, just like previous studies that create TimeGrad and CSDI. We envision our work as a revisiting study, unveiling new insights and previously neglected challenges in existing research, and providing a tool to encourage future research along these new paths.
> >
> > In conclusion, we hope that our responses satisfactorily address all your concerns. Should you have any further questions, we would be more than happy to engage in deeper discussions.

---

> ### Comment · Reviewer_kFUJ · 2023-11-21
>
> It seems from the latest response that the positioning of ProbTS is now as an empirical study, rather than a toolkit, so I shall now give my thoughts from the perspective of an empirical study (and therein lies my original critique -- this paper has split its focus between a library/toolkit, benchmark, and empirical study, failing to hit the mark on any).
>
> 1. Findings regarding datasets are not well-supported.
>
>     a. Exchange-S/L, Electricity-S/L, and possibly Traffic-S/L are exactly the same datasets, Solar has strong seasonality -- it is not clear how the data supports the conclusion that long term forecasting datasets have more trend/seasonality than short term forecasting datasets.
>
>     b. How does the JS div score lead to the conclusion that the datasets have more complex data distributions? Does this mean that all non-gaussian data distributions are more complex? What about a Student-T distribution head or Negative-binomial distribution head? These are simple modelling choices that can handle non-Gaussian data.
>
> 2. Findings regarding models do not constitute significant contributions.
>
>     a. "Probabilistic methods excel in modeling complex data distributions" -- probabilistic methods obviously are better than point forecasting methods at modelling distributions.
>
>     b. "Customized network architecture performs remarkably on long-term forecasting." -- similarly, these "customized network architectures" are methods proposed specifically for the LSTF setting, and naturally performs better on the LSTF setting.
>
>     c. "Probabilistic methods may produce poor point forecasts even with a superior CRPS score"  -- again, these findings are based on very shaky experimental design. How do we know probabilistic methods are worse than point forecasting methods on the NMAE metric due to it being a probabilistic method, rather than the architecture design? It has superior CRPS score due to the nature of the metric favouring probabilistic forecasts, but if you gave the more advanced architecture design a probabilistic head, would it have gotten a superior CRPS score? If you want to make such claims, the experiments have to be carefully designed, performing ablations on the different components. We cannot draw such conclusions from the benchmark presented.
>
>     d. "Customized network architecture on short-term forecasting remains underexplored" -- If you consider that the "customized network architectures" have mostly been proposed for the LSTF task, then this statement is straightforward and does not require experiments to arrive to

---

> > ### Author Response · Authors · 2023-11-22
> > **Response to the additional comment by Reviewer kFUJ (Part 1/3)**
> >
> > Thank you for your insightful and timely feedback. Your expertise in this domain is evident and we greatly appreciate the critical and visionary suggestions you have provided. In response to your comments, we would like to further discuss your specific concerns and update you on the steps we have taken to improve our paper based on your valuable input.
> >
> > Firstly, we acknowledge your critique regarding the paper's divided focus between a toolkit and an empirical study. Our intention for this paper is to position it as **a research-oriented tool that facilitates an intriguing revisit of two significant research branches in deep time-series forecasting**. Our goal is to reveal unique findings to the research community while concurrently offering a tool that facilitates easy benchmarking and further exploration in this area.
> >
> > Next, we include the responses to your specific concerns in two parts.
> >
> > **Response to "Findings regarding datasets are not well-supported."**
> >
> > a) Indeed, [Dataset]-S and [Dataset]-L are essentially identical datasets, but the versions used in the two branches slightly differ (as outlined in Table 2) and they utilize different forecasting horizons (S: short, L: long). Table 3 provides a comparison of trend and seasonality strengths at the dataset level. Furthermore, we have included a qualitative comparison of trend/seasonality strengths for various forecasting horizons using a sliding window to segment the whole time series, as depicted in Figures 2 & 3. As highlighted in "Section 4.1 Impacts of forecasting horizons", it is apparent that short time-series segments often display local dynamics (as seen in Figure 2) while long time-series segments demonstrate clear trending and prominent seasonality (as seen in Figure 3). Please note that these sliding windows are not selectively chosen. We have verified the results across all sliding windows and can confirm that these examples are representative of general cases.
> >
> > b) We would like to clarify that the JS div score, which measures how closely the value distribution aligns with a Gaussian distribution, is a universally applicable metric that effectively distinguishes between non-probabilistic methods and sophisticated probabilistic estimation methods. The reason behind this is that a non-probabilistic method, typically trained using MSE losses, can be seen as an equivalent probabilistic method when a Gaussian probabilistic head is added and learned through the Maximum a Posteriori (MAP) estimation from a Bayesian perspective ([https://en.wikipedia.org/wiki/Maximum_a_posteriori_estimation](https://en.wikipedia.org/wiki/Maximum_a_posteriori_estimation)). This equivalence implies that non-probabilistic methods inherently assume that their forecasts' distribution is a Gaussian centered around their point-level forecast values. As such, we consider our JS div score to be an appropriate measure of data complexity to discern non-probabilistic methods. Our results also validate its effectiveness in elucidating performance comparisons.
> >
> > Additionally, we concur with your observation that not all non-Gaussian data distributions are complex, and we want to clarify that we did not make such a claim. In reality, the underlying data distribution can be incredibly intricate and may not be easily captured by closed-form distributions such as the Student-T or Negative-binomial distribution. Using these closed-form distributions imposes a strong prior on the specific data scenarios. A more promising trend in recent years in probabilistic forecasting involves using a suitably parametrized network to learn data distributions without a fixed closed-form expression. This paper investigates pioneering studies in this trend, including GRU MAF, GRU NVP, Trans MAF, TimeGrad, and CSDI. For those complex data distributions that are difficult to characterize with a fixed rule, CRPS serves as a universal measure of how well the data distribution is captured ([https://en.wikipedia.org/wiki/Scoring_rule](https://en.wikipedia.org/wiki/Scoring_rule)).
> >
> > In conclusion, the JS div score utilized in our paper is specifically designed to distinguish between non-probabilistic methods, especially those incorporating custom neural architecture designs, and advanced probabilistic forecasting methods that do not adhere to a fixed data distribution format. For complex data distributions that are difficult to characterize mathematically, we advocate using CRPS as a universally applicable metric.

---

> > ### Author Response · Authors · 2023-11-22
> > **Response to the additional comment by Reviewer kFUJ (Part 2/3)**
> >
> > **Response to "Findings regarding models do not constitute significant contributions."**
> >
> > We apologize if the current exposition of our findings did not facilitate a clear understanding of the insights offered. Your insightful comments and questions have inspired us to refine our presentation for increased clarity.
> >
> > Upon careful review of your specific concerns, we find it essential to clarify two principles to fully elucidate our findings.
> >
> > - First, it's important to note that our findings, organized in five subsections in Section 4, are not standalone points. They are interrelated, providing a multi-faceted analysis of the comparison between the two research branches. Some of our statements in these sections may not have clearly conveyed these relationships, giving the impression of insignificant findings. We will address these concerns in the following responses.
> > - Second, it's important to underscore that the objective of this paper is to expose notable gaps between two significant research branches and to unveil new avenues for future exploration. Consequently, our focus is not on developing new architectures or probabilistic modules, but rather on benchmarking existing studies and conducting some ablation tests.
> >
> > With these principles in mind, we will now address your specific questions in greater detail.
> >
> > **In response to comment (a).** The assertion in Section 4.2, "Probabilistic methods excel in modeling complex data distributions," affirms the strengths of recently developed advanced probabilistic methods in short-term forecasting scenarios, especially when it comes to characterizing complex data distributions. Although long-term scenarios may also feature complex data distributions, often masked by prominent trend and seasonality, current probabilistic methods encounter challenges in these scenarios:
> >
> > - Most probabilistic methods opt for the auto-regressive decoding scheme, which probably suffers from substantial error propagation when dealing with prominent trends over long horizons. (This is detailed further in "Section 4.4 Autoregressive vs. Non-autoregressive decoding scheme")
> > - One non-autoregressive probabilistic method, CSDI, is significantly impeded by computational inefficiency and high memory requirements in long-term forecasting. (This is explored further in "Section 4.5 Computational Efficiency")
> >
> > When these findings are considered collectively, it becomes evident that there are numerous challenges when applying existing probabilistic methods to long-term forecasting. The underlying reason is that these methods are rarely evaluated in long-term scenarios. To the best of our knowledge, ours is among the first systematic studies on this aspect. Consequently, there is an abundance of research opportunities in this direction.
> >
> > **In response to comments (b) and (d).** Similarly, the statement in Section 4.3, "Customized network architecture performs remarkably on long-term forecasting," should be considered in conjunction with "Autoregressive models excel with strong seasonality but struggle with pronounced trends" from Section 4.4. Existing customized network architectures are specifically designed for long-term scenarios, which is why they all adopt a non-autoregressive decoding scheme. Does this mean we should disregard the autoregressive scheme in long-term scenarios? Our comparative studies suggest otherwise. We found that autoregressive models excel with strong seasonality even in long-term scenarios. If the effect of a large trend can be mitigated, autoregressive models can demonstrate impressive performance, as evidenced by GRU NVP and Time Grad on the Traffic dataset, shown in Table 5. Additionally, in response to Reviewer jfDV's request (weakness 3) for synthetic data experiments, we have further validated these findings.
> >
> > Given our findings, we perceive potential opportunities for autoregressive methods in long-term scenarios, provided an effective mechanism to alleviate error propagation is developed. Meanwhile, in contrast to the multitude of customized architectures developed for long-context scenarios driven by significant trend and seasonality, we underscore that "Customized network architecture for short-term forecasting remains underexplored" in Section 4.3. Specifically, when advanced probabilistic estimation techniques have effectively managed complex data distribution, what would happen if the research community dedicated more effort to integrating customized neural network designs? This is a direction we eagerly anticipate. However, it will be challenging, as the driving factors in short-term forecasting will not be trends or seasonality, which have already been explored by existing studies. We envision that once these open problems are clearly presented to the community, with more concerted research efforts, we can together make progress in this distinct direction.

---

> > ### Author Response · Authors · 2023-11-22
> > **Response to the additional comment by Reviewer kFUJ (Part 3/3)**
> >
> > **In response to comment (c).** "Probabilistic methods may produce poor point forecasts even with a superior CRPS score" refers to our findings detailed in Section 4.2 (supported by evidence in Table 5). We wish to clarify that it is the same model that exhibits the best CRPS score yet significantly larger NMAE compared to the model with the best NMAE (e.g., the CSDI model on ETTh2 datasets, 96 horizons). This implies that solely optimizing for CRPS may compromise the accuracy of point-level forecasts.
> >
> > In the context of our comparative study, the consistency in evaluation metrics highlights the urgent need for a more comprehensive evaluation using both probabilistic scores and point-level scores. Previous studies in the probabilistic forecasting often prioritize discussing probabilistic scores, leaving point-level metrics relegated to the appendix, as seen in studies like TimeGrad, or focusing solely on aggregated forecasts, referred to as CRPS-sum. Our findings, therefore, underscore the necessity of leveraging comprehensive evaluation metrics because in practice, we may require both accurate distributional estimations and precise point-level expected forecasts.
> >
> > Additionally, this finding inspires us to consider a hybrid learning strategy that excels in both probabilistic metrics like CRPS and non-probabilistic metrics like NMAE. This perspective could lead to entirely new design paradigms in time-series forecasting, offering multiple exploration opportunities. For instance, your suggestion of combining advanced neural architectures with a probabilistic head is a viable direction. However, the challenge lies in effectively integrating them. Other feasible solutions may involve deep fusions of architectural designs, probabilistic estimations, and optimization losses.
> >
> > In summary, the observation of inconsistency between CRPS and NMAE concretely supports the importance of this work, indicating that these two distinct research branches should be unified to foster new opportunities. The extensions stemming from this finding not only highlight the need for comprehensive evaluations but also reveal new research opportunities in these unified evaluations.
> >
> >
> > **Action Plans to Improve the Writing of this Paper**
> >
> > We sincerely appreciate your insightful and constructive feedback. Through the comprehensive analyses on these crucial topics prompted by your comments, we have formulated a strategy to improve the clarity and consistency of this paper. Your contributions have been instrumental in refining this paper, and we would like to express our gratitude once again for your valuable assistance.
> >
> > Specifically, we plan to move a large portion of Section 3, which details the main architecture of ProbTS, to the appendix. In its stead, we will include a discussion section in the main paper that consolidates all the necessary logic (including but not limited to what has been discussed above). This section aims to string together subsections 4.x and effectively communicate our unique findings, their implications for unmet challenges, and the intriguing potential for future research opportunities. This strategy will align more closely with the goal of this paper, revealing unique findings to the research community, while also providing a tool that facilitates benchmarking and promotes further exploration in this field.

---

### Official Review · Reviewer_aD4Z · 2023-11-01

**Soundness:** 3 good
**Presentation:** 3 good
**Contribution:** 3 good
**Rating:** 6
**Confidence:** 3

**Summary:**

The authors propose ProbTS, a toolkit for timeseries forecasting that implements a wide range of methods, and report a series of benchmarks that are thoroughly analyzed.

**Strengths:**

- This work presents an interesting analysis of various time-series forecasting methods, the authors do a great job bridging the gap between two different branches. The benchmarking of these methods across various datasets is valuable and the analysis is very insightful. The work reflects on the current strategies, and provides a unified view of current approaches and existing challenges, and would be invaluable for researchers working on these problems.
- I find the analyses incredibly insightful. The differences between the CRPS and NMAE metrics is interesting.
- The proposed toolbox is thorough and provides a unified framework for comparing various methods at an equal footing (same data pre-processing..). The most recent methods are implemented. This tool should be useful for researchers and could help bridge the gap between the two branches.

**Weaknesses:**

1. The datasets being studied are on the smaller scale. While these are the main benchmark datasets used in the field, comparing methods on datasets of varying sizes would be important. One might suggest that probabilistic methods excel with large amounts of data.
2. A noticeably absent aspect of time series is its multi-variate nature. Some methods like PatchTST for example independently process channels. Do different methods present limitations from not modeling the cross-channel interactions?

**Questions:**

1. Does ProbTS use standard hyperparameter tuning packages like raytune?
2. For transformer-based models, how were patch sizes determined?
3. How does model performance compare to performances reported in each method's respective paper? Were all the methods implemented in ProbTS reproduced successfully?

---

> ### Author Response · Authors · 2023-11-20
> **Response to Reviewer aD4Z (Part 1/3)**
>
> We are deeply grateful for your acknowledgment of the unique contributions made by this work. Your recognition of our analyses as both interesting and insightful is highly appreciated. In what follows, we will address your remaining concerns and questions specifically.
>
> **Response to Weaknesses 1**
>
> We appreciate your emphasis on the significance of comparing methods across datasets of varying sizes. Your viewpoint on this matter aligns with similar insights raised in Weakness 2 by Reviewer eUu7 and Question 2 by Reviewer AnhU. For further details, please refer to our responses to their queries. In the following, we present a comprehensive response to your question.
>
> First, we would like to clarify that the datasets currently included in ProbTS already cover a wide range of sizes. Below is a table summarizing the statistics of these datasets. As demonstrated, these datasets span a wide range of scales, from smaller sets containing 792 timesteps to larger ones with 69,680 timesteps, and in their dimensionality, ranging from low (7) to high (2000). We believe this wide array of datasets adequately substantiates our findings.
>
> Table 1. Statistics of datasets.
>
> | Dataset | # Var. | Timesteps |
> | --- | --- | --- |
> | ETTh1-L / ETTh1-L | 7 | 17, 420 |
> | ETTm1-L / ETTm2-L | 7 | 69, 680 |
> | Electricity-L | 321 | 26, 304 |
> | Traffic-L | 862 | 17, 544 |
> | Weather-L | 21 | 52, 696 |
> | Exchange-L | 8 | 7, 588 |
> | ILI-L | 7 | 966 |
> | Exchange-S | 8 | 6, 071 |
> | Solar-S | 137 | 7, 009 |
> | Electricity-S | 370 | 5, 833 |
> | Traffic-S | 963 | 4, 001 |
> | Wikipedia-S | 2000 | 792 |
>
>
> Furthermore, we have undertaken a quantitative analysis to discern the relationship between the data scale and the superiority of probabilistic methods. Our analysis revealed that dataset size does not play a pivotal role in influencing the design paradigm of the two branches under consideration. In our preliminary analysis, we examined the impact of data volume on performance and did not identify any clear correlation patterns, as demonstrated in the tables below.
>
> Table. The correlation coefficient between the data volume and the relative performance improvement of CRPS/NMAE compared to the baseline model (GRU).
>
> | Model | DLinear | PatchTST | GRU NVP | TimeGrad | CSDI |
> | --- | --- | --- | --- | --- | --- |
> | # Var. | 0.2422 / 0.2422 | -0.2676 / -0.2676 | -0.1856 / -0.2136 | -0.1665 / -0.1793 | -0.2315 / -0.2592 |
> | # Total timestep | -0.1422 / -0.1422 | 0.3821 / 0.3821 | 0.3072 / 0.3329 | 0.2860 / 0.2971 | 0.3542 / 0.3826 |
> | # Var. x Timestep | 0.0162 / 0.0162 | 0.0166 / 0.0166 | -0.0068 / -0.0011 | 0.0082 / 0.0117 | -0.0053 / -0.0133 |

---

> ### Author Response · Authors · 2023-11-20
> **Response to Reviewer aD4Z (Part 2/3)**
>
> **Response to Weaknesses 2**
>
> Thank you for bringing up the issue of modeling cross-channel interactions in time series forecasting. We have given this aspect careful thought, but as of now, we haven't discerned a clear pattern relating the modeling of cross-channel interactions to overall model performance. We have prepared a summary table illustrating how models from each branch handle the multivariate aspect, as detailed below.
>
> Table. Summary of how existing models handle multivariate time series.
>
> | Model | Research branch | Process channels independently |
> | --- | --- | --- |
> | N-BEATS | Customized neural architectures | √ |
> | N-HiTS | Customized neural architectures | √ |
> | Autoformer | Customized neural architectures | x |
> | Informer | Customized neural architectures | x |
> | LSTF-Linear | Customized neural architectures | x / √ |
> | PatchTST | Customized neural architectures | x / √ |
> | TimesNet | Customized neural architectures | x |
> | DeepAR | Probabilistic estimation | √ |
> | GP-copula | Probabilistic estimation | x |
> | LSTM NVP | Probabilistic estimation | x |
> | LSTM MAF | Probabilistic estimation | x |
> | Trans MAF | Probabilistic estimation | x |
> | TimeGrad | Probabilistic estimation | x |
> | CSDI | Probabilistic estimation | x |
> | SPD | Probabilistic estimation | x |
>
> A notable trend is the prevalent use of a channel-mixing approach in most studies. However, findings are diverse, with models like DLinear [1] and PatchTST [2] suggesting that processing channels independently can yield superior results, while others like CSDI [3] indicate that explicit modeling of cross-channel interactions offers significant advantages. This diversity points to the ongoing exploration of the impact of cross-channel interactions on forecasting performance.
>
> Additionally, we have consistently noted that as dimensionality escalates, the modeling of cross-channel interactions necessitates greater memory capacity, especially for methods based on probabilistic estimation.
>
> We sincerely value your insightful feedback, and we plan to incorporate a discussion on this topic in our revised manuscript. We hope this will stimulate more in-depth investigations into this significant aspect of time series forecasting.
>
> [1] Zeng, Ailing, et al. (2023). Are transformers effective for time series forecasting?." Proceedings of the AAAI. Vol. 37. No. 9.
>
> [2] Nie, Yuqi, et al. (2023). A time series is worth 64 words: Long-term forecasting with transformers. Proceedings of the ICLR.
>
> [3] Tashiro, Yusuke, et al. (2021). Csdi: Conditional score-based diffusion models for probabilistic time series imputation." Proceedings of the NeurIPS.

---

> ### Author Response · Authors · 2023-11-20
> **Response to Reviewer aD4Z (Part 3/3)**
>
> **Response to Question 1**
>
> As outlined in Appendix C.1, we adopted an exhaustive grid search approach for model tuning. We acknowledge your valuable suggestion to employ standard hyperparameter tuning packages. In future iterations, we will investigate incorporating such packages to improve the efficiency of parameter tuning, particularly when introducing additional baselines.
>
> **Response to Question 2**
>
> To ensure a fair comparison, we set the hyperparameters via thorough standard tuning procedures. For those models and datasets where hyperparameters were previously provided, we reused the reported patch size that had been demonstrated to deliver optimal results, a finding that our reproduction also validates. When it came to short-term forecasting and patch size settings were not available, we conducted a search for the ideal patch sizes within a range of 4 to 8. This was done considering the shorter lookback window and forecasting horizon.
>
> **Response to Question 3**
>
> In most instances, where detailed reproducible specifications were provided (for both datasets and models), we successfully reproduced results by directly applying or slightly adjusting the reported hyperparameters. For those scenarios without clear guidance, such as evaluating probabilistic methods in long-term forecasting scenarios, we conducted a unified grid search over some critical hyperparameters on the validation set to determine the final experimental setups.
>
> However, it's important to approach comparisons of our results with those in the original papers cautiously due to subtle differences in metric calculations. Some methods denormalized their experimental results before evaluation [1,2,3], while others calculated the metric scores directly without denormalization [4,5], creating a challenge for direct comparison. To address this, our toolkit preserves both denormalized and non-denormalized results, facilitating a unified evaluation and comparison. In our main paper, we report the evaluation metrics calculated at the original value scale (after denormalization).
>
> We made every effort to ensure a fair evaluation of all baselines, and we plan to open-source all the code and configuration files. This commitment to fairness underpins the insights we derived, which form the primary contribution of our paper. We appreciate your understanding of these complexities and welcome further suggestions to enhance the transparency and robustness of our comparisons.
>
> [1] Rasul, Kashif, et al. (2021). Multivariate probabilistic time series forecasting via conditioned normalizing flows. Proceedings of the ICLR.
>
> [2] Rasul, Kashif, et al. (2021). Autoregressive denoising diffusion models for multivariate probabilistic time series forecasting. Proceedings of the ICML.
>
> [3] Tashiro, Yusuke, et al. (2021). CSDI: Conditional score-based diffusion models for probabilistic time series imputation. Proceedings of the NeurIPS.
>
> [4] Wu, Haixu, et al. (2022). Timesnet: Temporal 2d-variation modeling for general time series analysis. Proceedings of the ICLR.
>
> [5] Nie, Yuqi, et al. (2022). A time series is worth 64 words: Long-term forecasting with transformers. Proceedings of the ICLR.

---

### Official Review · Reviewer_JfDV · 2023-11-07

**Soundness:** 3 good
**Presentation:** 4 excellent
**Contribution:** 3 good
**Rating:** 8
**Confidence:** 4

**Summary:**

This paper introduces ProbTS, a novel toolkit aimed at bridging the gap between two prominent research branches in time-series forecasting: one focused on customized neural network architectures and the other on advanced probabilistic estimations. The paper highlights key insights from the toolkit's analysis, revealing that long-term forecasting scenarios often exhibit strong trending and seasonality patterns, while short-term scenarios have more complex data distributions. It also identifies the strengths and weaknesses of different methodological focuses, showing that probabilistic forecasting excels in modeling data distributions, but may produce poor point forecasts. Additionally, the autoregressive decoding scheme is effective in cases with strong seasonality but struggles with pronounced trending, while the non-autoregressive scheme is preferred for long-term forecasting. The paper concludes by emphasizing the potential of combining these research branches to revolutionize time-series forecasting and anticipates that ProbTS will catalyze groundbreaking research in the field.

**Strengths:**

The paper possesses several strengths. Firstly, it is well-written, displaying a high level of clarity and organization. The insights provided are undeniably valuable, shedding light on the challenging questions arising from the divergence in data scenarios, methodological approaches, and decoding schemes within the realm of time-series forecasting. The paper effectively highlights the significant gap in the existing literature, where no prior solution has successfully bridged the divide between these two distinct research branches. This emphasis on addressing an unexplored area of research stimulates further groundbreaking work in the field. Moreover, the sharing of the ProbTS toolkit included in the paper will benefit the research community, offering a practical resource to help researchers understand and effectively handle these complex issues, ultimately fostering collaboration and collective progress in the field.

**Weaknesses:**

While this paper offers valuable insights and contributions, there are a few areas where it could be improved. Firstly, while the ProbTS toolkit is undoubtedly a valuable resource, for me, the insights presented in the paper are very informative, and I believe that placing a stronger emphasis on these insights would have been greatly beneficial.

Additionally, the paper could benefit from more extensive discussions on other critical characteristics of time-series forecasting, such as dimensionality, data length, or the volume of training data. These factors can significantly impact forecasting performance, and a deeper exploration of their effects would have been highly informative.

Moreover, the insights presented in the paper could have been more rigorously developed and supported. The use of synthetic datasets and controlled experiments could have strengthened the empirical evidence, particularly since the datasets used in the analysis exhibit diverse characteristics that might confound the results.

Lastly, a minor point of improvement lies in the Contributions section of the Introduction, where the term CRPS is mentioned before its definition. Providing a definition before using the abbreviation would enhance the clarity of the paper.

**Questions:**

See Weaknesses.

---

> ### Author Response · Authors · 2023-11-20
> **Response to Reviewer JfDV**
>
> Thank you for your comprehensive review and constructive feedback. We also particularly appreciate your recognition of our work. We hope that the following responses will address your concerns and contribute to enhancing the completeness and rigor of this paper.
>
> **Response to Weaknesses 1**
>
> We greatly appreciate your suggestion and welcome any specific recommendations you may have to further highlight these insights. In the submitted manuscript, we briefly outlined these insights in the Introduction section and provided detailed discussions in Section 4. The key insights of each paragraph have been accentuated by bolding them and placing them at the beginning of each section. In the upcoming revision, we plan to improve our writing to ensure that the core insights of each paragraph are conveyed more effectively to the reader.
>
> **Response to Weaknesses 2**
>
> We appreciate your suggestion about the need for a more extensive discussion on the critical characteristics of time-series forecasting, such as dimensionality, data length, and the volume of training data. We did examine these factors in our preliminary studies, but they did not profoundly influence the design paradigm of the two branches under consideration.
>
> We did not observe a significant correlation between these factors and model performance, as indicated in the tables below. As these findings did not provide substantial insights, we chose not to include them in the paper. Nevertheless, we believe the ProbTS Toolkit is well-equipped to facilitate more in-depth analyses of these characteristics in future research if needed.
>
> Table 1. The correlation coefficient between the data volume and the relative performance improvement of CRPS/NMAE compared to the baseline model (GRU).
>
> | Model | DLinear | PatchTST | GRU NVP | TimeGrad | CSDI |
> | --- | --- | --- | --- | --- | --- |
> | # Var. | 0.2422 / 0.2422 | -0.2676 / -0.2676 | -0.1856 / -0.2136 | -0.1665 / -0.1793 | -0.2315 / -0.2592 |
> | # Total timestep | -0.1422 / -0.1422 | 0.3821 / 0.3821 | 0.3072 / 0.3329 | 0.2860 / 0.2971 | 0.3542 / 0.3826 |
> | # Var. x Timestep | 0.0162 / 0.0162 | 0.0166 / 0.0166 | -0.0068 / -0.0011 | 0.0082 / 0.0117 | -0.0053 / -0.0133 |
>
>
> **Response to Weaknesses 3**
>
> We appreciate your constructive feedback. We concur that controlled experiments and the use of synthetic datasets could provide more robust empirical evidence. Consequently, in response to your suggestion, we have generated synthetic datasets including a baseline dataset, one with pronounced trends, another with strong seasonality, and a dataset displaying complex data distribution. Each series within these datasets was created by combining trend, seasonality, noise, and anomaly components with controlled characteristics. The details are provided in the table below.
>
> Table 2. Quantitative assessment of intrinsic characteristics for synthetic datasets.
>
> | Dataset/Horizon | Trend | Seasonality | JS Div. |
> | --- | --- | --- | --- |
> | Normal | 0.105 | 0.302 | 0.261 |
> | Strong Trend | 0.554 | 0.302 | 0.248 |
> | Strong Seasonality | 0.105 | 0.791 | 0.272 |
> | Complex data distribution | 0.064 | 0.190 | 0.469 |
>
> Subsequent experiments conducted on these synthetic datasets, utilizing representative models, validated our empirical findings from other datasets examined with ProbTS. Noteworthy observations include the declining performance of autoregressive decoding models such as TimeGrad in the presence of increasing trends, an enhancement in performance for models employing autoregressive decoding with intensifying seasonality, and the competitive performance exhibited by probabilistic methods like CSDI when dealing with more complex data distributions.
>
> Table 3. Results on synthetic datasets. The look-back window and forecasting horizon are 30.
>
> | Model | DLinear | PatchTST | TimeGrad | CSDI |
> | --- | --- | --- | --- | --- |
> | Normal (CRPS) | 0.013  | 0.012 | 0.024 | 0.013 |
> | Normal (NMAE) | 0.013  | 0.012 | 0.032 | 0.014 |
> | Strong Trend (CRPS) | 0.001 | 0.001 | 0.042 | 0.010 |
> | Strong Trend (NMAE) | 0.001 | 0.001 | 0.048 | 0.007 |
> | Strong Seasonality (CRPS) | 0.014 | 0.012 | 0.022 | 0.020 |
> | Strong Seasonality (NMAE) | 0.014 | 0.012 | 0.028 | 0.027 |
> | Complex data distribution (CRPS) | 0.301 | 0.275 | 0.283 | 0.269 |
> | Complex data distribution (NMAE) | 0.301 | 0.275 | 0.338 | 0.301 |
>
> These observations further reinforce our findings, and we plan to incorporate these synthetic experiments and analyses in future revisions of our work. In addition, we aim to include the code for generating synthetic datasets within the ProbTS toolkit to encourage and facilitate further exploration of this area by the wider research community. We greatly appreciate your valuable feedback!
>
> **Response to Weaknesses 4**
>
> Thank you for your suggestion. We will take care to ensure that all abbreviations, including CRPS, are clearly defined prior to their first use in the revised version of the paper.

---

> > ### Comment · Reviewer_JfDV · 2023-11-21
> >
> > Thank you for your response to my review and for providing the additional results. I appreciate the effort you've invested in addressing the comments raised during the review process.
> >
> > I would like to commend you on the quality of your work. Having carefully considered your responses and the supplementary materials, I am confident that the paper has maintained its high standard. I believe the rating I assigned in my initial review accurately reflects the level of your research, and I see no need for adjustments.

---

### Official Review · Reviewer_AnhU · 2023-11-07

**Soundness:** 3 good
**Presentation:** 3 good
**Contribution:** 2 fair
**Rating:** 5
**Confidence:** 4

**Summary:**

The authors propose a toolkit to evaluate time-series forecasting methods on various datasets. They observe that there are two main branches: long-term forecasting where data revlease strong trends and seasonality patterns, and a second branch oriented towards short-term forecasting

Highlighting that different data characteristics and forecasting horizons prefer different design

Long - term forecasting : specializing in neural network architecture design with various inductive biases, restricting themselves to point-forecasts
Short - lean towards conventional neural network designs

**Strengths:**

- Authors implement quite a few models which are evaluated on on the datasets
- The framework provides a standardized way of evaluating methods

**Weaknesses:**

There are multiple time-series survey/benchmark papers in the literature for forecasting which emphasize standardization across datasets [1], others that emphasize architectural studies [2] and [3] which classifies time-series forecasting methods along the same direction as this work.

It’s not clear where the authors proposed framework fits amongst these previous studies on time-series forecasting, it looks like another way of characterizing time-series forecasting models which is partially covered in [3]

[1] Godahewa, Rakshitha, et al. "Monash time series forecasting archive." arXiv preprint arXiv:2105.06643 (2021).
[2] Elsayed, S., et al. "Do we really need deep learning models for time series forecasting? arXiv 2021." arXiv preprint arXiv:2101.02118.
[3] Januschowski, Tim, et al. "Criteria for classifying forecasting methods." International Journal of Forecasting 36.1 (2020): 167-177.

**Questions:**

- Why don’t authors compare with simpler methods such as XGBoost (with hand crafted features?)
   - it's quite hard to beat this baseline on the datasets that were used in this paper.
- The datasets used are quite small, I'm curious if these findings hold if we increase dataset size
- Hyperparameters and preprocessing steps used for these datasets could dramatically effect model performance. Were these tuned individually for each of the methods? And why is this not included as part of the text
- What is the guiding mechanism for determining whether a dataset suits the short-forecast or long-forecast category? Is it simply the forecasting window? Or rather intrinsic property to the dataset
- I believe although initially the paper tries to consider both model/and data aspects of the time-series forecasting domain it fails to provide concrete guidance on how one effects the other, i.e. a quantifiable way of delineating which approach should be taken

---

> ### Author Response · Authors · 2023-11-20
> **Response to Reviewer AnhU (Part 1/3)**
>
> Thank you for your insightful comments and for recognizing some unique contributions of our work. We would like to further clarify our motivation and the core position of this work to enhance your understanding. We regret any misunderstanding and would like to assure you that this work is not just about combining long-term and short-term forecasting.
>
> Firstly, the primary motivation of this work is to **unify two equally important yet independently developed research branches in deep time-series forecasting**:
>
> - One branch focuses on customized neural architecture designs but only provides point-level forecasts.
> - The other branch specializes in probabilistic estimations for distributional forecasts but rarely includes architecture designs.
>
> Secondly, during this unification research, we identified several critical aspects that need to be aligned, including:
>
> - The role of different data characteristics (trend, seasonality, complexity of data distribution) and forecasting setups (long-term vs short-term) in influencing the methodological designs and choices.
> - The existence of several unexplored scenarios and overlooked challenges in the literature due to different methodological designs in rarely visited data scenarios. The significant methodological aspects we've summarized include:
>     - Probabilistic vs non-probabilistic forecasting paradigms and evaluation metrics.
>     - Customized vs general neural architecture designs.
>     - Autoregressive vs non-autoregressive decoding schemes.
>
> Thirdly, after conducting experimental comparisons and analyses of these methodological designs across diverse data scenarios, we present our unique findings and insights on the strengths and weaknesses of different methods. We believe the challenges we uncovered and the new opportunities we identified will pave the way for future research aimed at better combining the benefits of neural architecture designs and probabilistic estimation abilities.
>
> We hope the above clarification of our paper's logic can help to reinforce your understanding of the unique contributions of our work. With this context, we will proceed to address your specific concerns in more detail.
>
> **Response to Weaknesses**
>
> We appreciate your reference to these important studies on time-series benchmarking. We plan to incorporate a discussion on these works in our revised paper. However, it's crucial to highlight the fundamental differences between our work, ProbTS, and these studies.
>
> In brief, [1] provides a broad range of datasets and benchmarks a multitude of forecasting methods, [2] primarily compares gradient boosted decision trees with deep learning methods, and [3] aims to categorize existing methods in different aspects, appearing more like a literature review. Specifically, neither [1] nor [2] consider probabilistic forecasting, while [3] focuses on fine-grained classifications of forecasting methods beyond the simple "machine learning" or "statistical" dichotomy.
>
> It's noteworthy that these studies were published at least two years ago. Given the rapid development in this field, our work is unique in its recognition of the need to **unify the research in customized neural architectures and advanced probabilistic estimation abilities**, both of which are vital components in deep time-series forecasting but have evolved independently in recent literature. In this context, our work reveals several unique insights and findings, such as:
>
> - The distinction between long-term and short-term forecasting, along with different data characteristics.
> - Probabilistic vs non-probabilistic forecasting paradigms and evaluation metrics.
> - Customized vs general neural architecture designs.
> - Autoregressive vs non-autoregressive decoding schemes.
>
> In summary, our work is not merely another benchmarking study nor a comprehensive review covering all aspects of time-series forecasting. We view our work as a critical step in bridging the prominent yet divergent branches in contemporary research on time-series forecasting.
>
> [1] Godahewa, Rakshitha, et al. (2021). Monash time series forecasting archive. *arXiv preprint arXiv:2105.06643*.
>
> [2] Elsayed, S., et al. (2021). Do we really need deep learning models for time series forecasting?. *arXiv preprint arXiv:2101.02118*.
>
> [3] Januschowski, Tim, et al. (2020). Criteria for classifying forecasting methods. *International Journal of Forecasting*, *36*(1), 167-177.

---

> ### Author Response · Authors · 2023-11-20
> **Response to Reviewer AnhU (Part 2/3)**
>
> **Response to Question 1**
>
> We are grateful for your thoughtful suggestion to include traditional tree-based methods, such as XGBoost, in our comparison. Following the implementation provided by [1], we found that XGBoost indeed exhibits competitive performance, even surpassing neural network (NN) models in certain scenarios.
>
> However, for datasets with more complex data distributions such as Solar and Electricity, advanced probabilistic estimation methods demonstrated a significant advantage over both traditional learning methods and point estimation methods. This underscores the versatility and strength of these advanced probabilistic methods in handling complex forecasting scenarios.
>
> Despite the competitive performance of XGBoost, one of its limitations is its reliance on handcrafted features, which contradicts the recent trend toward automatic representation learning inherent in NN models. This is one of the key reasons why, in line with many recent studies, we've chosen to focus on deep time-series forecasting.
>
> Furthermore, it's important to note that both tree-based methods, such as XGBoost, and NN models necessitate extensive hyper-parameter tuning. Despite our rigorous hyper-parameter tuning efforts, we did not achieve satisfactory performance with XGBoost on the Wikipedia dataset. This outcome may suggest that additional feature engineering is required for this specific dataset. However, considering the time constraints and the substantial efforts involved in preparing responses, we have decided to exclude this dataset from the subsequent table.
>
> Your insightful suggestion has undoubtedly enriched our study. By demonstrating the strengths of both XGBoost and NN models, we've provided a more comprehensive view of the current landscape of time-series forecasting. We believe this expanded perspective will enhance the overall depth and value of our paper. We greatly appreciate your contribution to this improvement.
>
> | Model (Metric) | Category | Solar | Electricity | Traffic | Exchange |
> | :--- | :--- | :---: | :---: | :---: | :---: |
> | XGBoost (CRPS) | GBDT | 0.599 | 0.074 | 0.196 | 0.011 |
> | XGBoost (NMAE) | GBDT | 0.599 | 0.074 | 0.196 | 0.011 |
> | NLinear (CRPS) | Customized NN | 0.560 | 0.083 | 0.233 | 0.010 |
> | NLinear (NMAE) | Customized NN | 0.560 | 0.083 | 0.233 | 0.010 |
> | PatchTST (CRPS) | Customized NN | 0.496 | 0.076 | 0.202 | 0.010 |
> | PatchTST (NMAE) | Customized NN | 0.496 | 0.076 | 0.202 | 0.010 |
> | TimeGrad (NMAE) | Probabilistic estimation | 0.359 | 0.052 | 0.164 | 0.011 |
> | TimeGrad (NMAE) | Probabilistic estimation | 0.445 | 0.067 | 0.201 | 0.014 |
> | CSDI (CRPS) | Probabilistic estimation | 0.366 | 0.050 | 0.146 | 0.008 |
> | CSDI (NMAE) | Probabilistic estimation | 0.484 | 0.065 | 0.176 | 0.011 |
>
> [1] Elsayed, S., et al. (2021). Do we really need deep learning models for time series forecasting?. *arXiv preprint arXiv:2101.02118*.
>
>
> **Response to Question 2**
>
> We appreciate your insightful question. ProbTS has been designed and tested on a diverse range of datasets, as illustrated in the table below. These datasets vary significantly in scale, ranging from small (792) to large (69,680), and in dimensionality, from low (7) to high (2000). We believe this broad spectrum of datasets adequately validates our findings.
>
> Table 1. Statistics of datasets.
>
> | Dataset | # Var. | Timesteps |
> | --- | --- | --- |
> | ETTh1-L / ETTh1-L | 7 | 17, 420 |
> | ETTm1-L / ETTm2-L | 7 | 69, 680 |
> | Electricity-L | 321 | 26, 304 |
> | Traffic-L | 862 | 17, 544 |
> | Weather-L | 21 | 52, 696 |
> | Exchange-L | 8 | 7, 588 |
> | ILI-L | 7 | 966 |
> | Exchange-S | 8 | 6, 071 |
> | Solar-S | 137 | 7, 009 |
> | Electricity-S | 370 | 5, 833 |
> | Traffic-S | 963 | 4, 001 |
> | Wikipedia-S | 2000 | 792 |
>
> In addition, we specifically investigated the impact of dataset size on model performance. Our analysis, depicted in Table 3, did not reveal a clear correlation between dataset size and model performance. Consequently, this aspect was not emphasized in our paper.
>
> Table 2. The correlation coefficient between the data volume and the relative performance improvement of CRPS/NMAE compared to the baseline model (GRU).
>
> | Model | DLinear | PatchTST | GRU NVP | TimeGrad | CSDI |
> | --- | --- | --- | --- | --- | --- |
> | # Var. | 0.2422 / 0.2422 | -0.2676 / -0.2676 | -0.1856 / -0.2136 | -0.1665 / -0.1793 | -0.2315 / -0.2592 |
> | # Total timestep | -0.1422 / -0.1422 | 0.3821 / 0.3821 | 0.3072 / 0.3329 | 0.2860 / 0.2971 | 0.3542 / 0.3826 |
> | # Var. x Timestep | 0.0162 / 0.0162 | 0.0166 / 0.0166 | -0.0068 / -0.0011 | 0.0082 / 0.0117 | -0.0053 / -0.0133 |

---

> ### Author Response · Authors · 2023-11-20
> **Response to Reviewer AnhU (Part 3/3)**
>
> **Response to Question 3**
>
> Your question is indeed pertinent. As specified in Appendix C.1, we carried out an extensive grid search for models, tuning hyperparameters individually for each method. Given the large number of models and the variance in hyperparameters across different datasets and forecasting scenarios, we chose not to include a detailed account of parameter configurations in the main text of the paper. However, we have made the specific hyperparameter configurations available in an open-source format along with the code.
>
> In response to your valuable suggestion, we will include the key hyperparameter settings in the upcoming revision to offer readers a more accessible reference point. Thank you for raising this important issue.
>
>
> **Response to Question 4**
>
> We appreciate your thoughtful question. The categorization of datasets into short-term or long-term forecasting is primarily influenced by practical necessities in various domains, rather than our personal determination of which dataset is best suited for a particular forecast duration. For instance, certain datasets like Wikipedia are commonly utilized for short-term forecasting. Others, such as Weather-L and ETTm1-L / ETTm2-L, are typically employed for long-term forecasting. Some datasets, like Electricity and Traffic, are applicable to both short-term and long-term forecasting.
>
> In this paper, our aim is not just to distinguish between short-term and long-term forecasting or varying domains. Instead, we strive to identify inherent dataset characteristics that may guide different methodological designs. The key characteristics we have identified include the strength of trending, the strength of seasonality, and the complexity of data distribution. These traits are a collective result of the length of forecasting horizons and the specific data domains.
>
> Our exploration into the interplay between data characteristics and model designs prompts us to envision future research that would conduct more comprehensive evaluations. These evaluations would span across different domains and forecasting horizons (short versus long), and would utilize a variety of evaluation metrics, both at the distributional and point level. We believe that ProbTS is well-equipped to facilitate this kind of comprehensive analysis. We value your feedback and will strive to clarify our presentation further in the next revision.
>
> **Response to Question 5**
>
> Thank you for your thoughtful question. We appreciate the opportunity to elucidate the synergistic relationship between model and data aspects in our research.
>
> Firstly, our work does indeed shed light on the interplay between these two aspects. We have found that specific data characteristics favor certain methodological designs, leading to a convergence in different studies, a phenomenon we have outlined in our paper.
>
> Secondly, we have evaluated models in data scenarios that have been scarcely studied in the past. This approach has allowed us to uncover overlooked challenges and unresolved problems in the field, thus contributing to a more holistic understanding of the topic.
>
> For example, our work includes the following findings.
>
> - Probabilistic methods demonstrate proficiency in modeling complex data distributions, even though they may produce poor point forecasts even with a superior CRPS score (Section 4.2).
> - Customized network architecture shows remarkable performance in long-term forecasting, yet it remains an underexplored area in short-term forecasting scenarios (Sections 4.3).
> - Autoregressive models excel in handling strong seasonality but face challenges with pronounced trends, while both autoregressive and non-autoregressive decoding schemes perform comparably well in short-term forecasting (Section 4.4).
>
> Lastly, the intricate interplay between data and model is a persistent theme in machine learning research. We have developed ProbTS in the hope that it will facilitate a deeper exploration of this dynamic in the time-series domain. ProbTS achieves this by providing comprehensive evaluation scenarios and metrics, and by unifying essential modeling components.
>
> We hope this clarifies the intent and findings of our work. Thank you for prompting this valuable discussion.

---

### Official Review · Reviewer_eUu7 · 2023-11-07

**Soundness:** 3 good
**Presentation:** 3 good
**Contribution:** 3 good
**Rating:** 8
**Confidence:** 5

**Summary:**

The paper presents a novel framework for joint training and evaluation of deep time series models on a multitude of datasets available in the literature. The key feature of the proposed approach is the ability to combine and evaluate probabilistic and non-probabilistic methods in one place.

**Strengths:**

- Work on developing a unified deep learning framework for time-series forecasting is very much appreciated
- The case study reveals interesting insights comparing short term and long term and probabilistic and non-probabilistic models

**Weaknesses:**

- ProbTS does not include any naive and statisitcal models (e.g. ETS). The lack of good functioning naive/statistical models for probabilistic forecasting is actually a significant gap in the modern deep learning literature. Could you please include a few methods from this area as baselines in the proposed framework?
- The benchmark contains many datasets, however key datasets that have been instrumental in designing some of the current architectures are missing. Can you include M4, M5, TOURISM?
- Most datasets included in the benchmark are small-scale. For the purpose of studying model scaling and ability to model complex distributions, it feels urgent that large scale time series datasets are included in modern benchmarks. In this context, I can think of FRED from https://arxiv.org/abs/2002.02887

**Questions:**

- Does your benchmark support zero-shot/few-shot/transfer learning training/testing, pretrained models, model zoo? If not, is it easy to extend it to this scenario? Can you touch on this topic in the paper?
- Does the framework support datasets that don't fit in RAM, what is the mechanism for dataset storage and loading? How do you deal with the licenses of original datasets?
- I included a number of questions and concerns and will be very happy to revise my score accordingly if all of them are addressed meticulously.

---

> ### Author Response · Authors · 2023-11-20
> **Response to Reviewer eUu7 (Part 1/3)**
>
> Thank you for your thorough review. We are particularly grateful for your recognition of the key strengths of our paper, specifically our unification of probabilistic and non-probabilistic methods to reveal novel insights, identify current challenges, and suggest potential directions for future research. In response to your additional queries, we trust that the following explanations will satisfactorily address your concerns.
>
>
> **Response to Weaknesses 1**
>
> Thank you for this insightful suggestion. As per your recommendation, we have incorporated a few classical statistical methods such as ARIMA and ETS into the ProbTS framework. This addition serves to further solidify and enrich our empirical research. We invite you to review the table below which compares these baseline methods with state-of-the-art deep learning techniques.
>
> | Model (Metric) | Solar | Electricity | Traffic | Wikipedia | Exchange |
> | :--- | :---: | :---: | :---: | :---: | :---: |
> | ARIMA (CRPS) | 1.000 | 0.164 | 0.461 | 0.348 | 0.009 |
> | ARIMA (NMAE) | 1.000 | 0.164 | 0.461 | 0.348 | 0.009 |
> | ETS (CRPS) | 0.580 | 0.121 | 0.413 | 0.685 | 0.011 |
> | ETS (NMAE) | 0.580 | 0.121 | 0.413 | 0.685 | 0.011 |
> | ETS-prob (CRPS) | 0.795 | 0.123 | 0.38 | 0.625 | 0.008 |
> | ETS-prob (NMAE) | 0.695 | 0.129 | 0.433 | 0.697 | 0.011 |
> | PatchTST (CRPS) | 0.496 | 0.076 | 0.202 | 0.257 | 0.010 |
> | PatchTST (NMAE) | 0.496 | 0.076 | 0.202 | 0.257 | 0.010 |
> | CSDI (CRPS) | 0.366 | 0.050 | 0.146 | 0.219 | 0.008 |
> | CSDI (NMAE) | 0.484 | 0.065 | 0.176 | 0.259 | 0.011 |
>
> As can be observed, deep learning methods significantly outperform these simple statistical baselines. This highlights the crucial role of capturing non-linear dependencies in generating accurate forecasts. Such comparisons underscore the necessity of further research to enhance deep time-series forecasting, which is one of the primary goals of our paper. Additionally, we have noted that ARIMA and ETS have distinct performance tendencies depending on specific data characteristics. For instance, the ARIMA method struggles with datasets like 'Solar' which exhibit weak trending and strong seasonality, whereas the ETS model shows better adaptability. Conversely, in cases of strong trending and weak seasonality, as seen in the 'Wikipedia' dataset, the ARIMA model significantly outperforms the ETS model.
>
> [1] Asteriou, Dimitros; Hall, Stephen G. (2011). "ARIMA Models and the Box-Jenkins Methodology". *Applied Econometrics* (Seconded.). Palgrave MacMillan. pp.265-286.[ISBN](https://en.wikipedia.org/wiki/ISBN_(identifier))[978-0-230-27182-1](https://en.wikipedia.org/wiki/Special:BookSources/978-0-230-27182-1).
>
> [2] Hyndman, Rob J., and Athanasopoulos, George. *Forecasting: principles and practice*, 3rd edition, OTexts, 2021.[https://otexts.com/fpp3/expsmooth.html](https://otexts.com/fpp3/expsmooth.html)
>
> [3] [https://www.statsmodels.org/dev/examples/notebooks/generated/ets.html](https://www.statsmodels.org/dev/examples/notebooks/generated/ets.html)

---

> ### Author Response · Authors · 2023-11-20
> **Response to Reviewer eUu7 (Part 2/3)**
>
> **Response to Weaknesses 2**
>
> Thank you for your suggestion to incorporate the M4, M5, and TOURISM datasets, which are indeed key datasets for univariate time-series forecasting. In light of your advice, we have analyzed these datasets and conducted experiments comparing different forecasting methods.
>
> Our findings indicate that the characteristics of these new datasets do not extend beyond the scope of the existing datasets used in this paper. Furthermore, the additional tests run on these new datasets further reinforce the major findings and conclusions of our paper. We provide more detailed information about these two points below for your convenience.
>
> Table 1 presents the comparison of the new datasets (M4, M5, TOURISM) with the existing datasets. As per the data analysis strategy detailed in our paper, we focus on three key aspects: the strength of trending, the strength of seasonality, and the complexity of data distribution. From the table, it is evident that these new datasets do not introduce particularly unique characteristics, with the exception of M4-Daily, which may exhibit fewer seasonal patterns. The main difference between the new datasets and the existing ones is that the new datasets are all designed for univariate time-series forecasting, while the datasets in our study primarily target multivariate time-series forecasting. Thanks to your constructive suggestion, we now have a broader spectrum covering both univariate and multivariate scenarios, thereby reinforcing our findings.
>
> Table 1. Quantitative assessment of the intrinsic characteristics of the datasets reported in our paper. The JS Div. denotes Jensen-Shannon divergence, where a lower score indicates closer approximations to a Gaussian distribution.
>
> | Dataset | Trend | Seasonality | JS Div. |
> | :--- | :---: | :---: | :---: |
> | Exchange | 0.9982 | 0.1256 | 0.2967 |
> | Solar | 0.1688 | 0.8592 | 0.5004 |
> | Electricity | 0.6443 | 0.8323 | 0.3579 |
> | Traffic | 0.2880 | 0.6656 | 0.2991 |
> | Wikipedia | 0.5253 | 0.2234 | 0.2751 |
>
> Table 2. Quantitative assessment of the intrinsic characteristics of the univariate datasets.
>
> | Dataset | Trend | Seasonality | JS Div. |
> | :--- | :---: | :---: | :---: |
> | M4-Weekly | 0.7677 | 0.3401 | 0.5106 |
> | M4-Daily | 0.9808 | 0.0467 | 0.4916 |
> | M5 | 0.3443 | 0.2480 | 0.6011 |
> | TOURISM-Monthly | 0.7979 | 0.6826 | 0.3291 |
>
> Table 3 includes the experimental results of some representative methods. These findings align with our initial observations discussed in the paper. For example, methods that specialize in probabilistic estimation, such as GRU NVP and TimeGrad, demonstrate superior performance on datasets with complex distributions like M4-Weekly and M5. On the other hand, on the TOURISM-Monthly dataset, which has a relatively simple data distribution, baseline methods equipped with a simple point forecaster, like DLinear and PatchTST, also perform well. Additionally, both autoregressive and non-autoregressive decoding schemes show comparable performance in short-term forecasting.
>
> Table 3. Results on M4, M5, and TOURISM datasets. We utilize a lookback window of 3H, with 'H' denoting the forecasting horizon.
>
> | Dataset (Metric) | DLinear | PatchTST | GRU NVP |
> | :--- | :---: | :---: | :---: |
> | M4-Weekly (CRPS) | 0.081  | 0.089 | 0.066 |
> | M4-Weekly (NMAE) | 0.081  | 0.089  | 0.077 |
> | M4-Daily (CRPS) | 0.034  | 0.035  | 0.030 |
> | M4-Daily (NMAE) | 0.034 | 0.035 | 0.038  |
> | M5 (CRPS) | 0.891 | 0.898 | 0.679 |
> | M5 (NMAE) | 0.891 | 0.898 | 0.864 |
> | TOURISM-Monthly (CRPS) | 0.168  | 0.136 | 0.171  |
> | TOURISM-Monthly (NMAE) | 0.168 | 0.136  | 0.223 |

---

> ### Author Response · Authors · 2023-11-20
> **Response to Reviewer eUu7 (Part 3/3)**
>
> **Response to Weaknesses 3**
>
> Thank you for your insightful observation regarding the scale of the datasets used. We concur with your assertion that the inclusion of large-scale datasets is crucial for future research development. As per your recommendation, we plan to progressively incorporate significant, distinctive, and large-scale datasets into ProbTS, starting with the FRED dataset collection.
>
> In the meantime, we would like to underscore that the diversity of data is another crucial aspect that warrants thorough investigation, which is the primary focus of this paper. Evaluating learning ability across different data scales, variate numbers, and domains with a wide scope of data characteristics is of great importance. Therefore, this paper is dedicated to evaluating diverse data characteristics and conducting comprehensive analyses to shed light on overlooked challenges and unresolved issues.
>
> The datasets included in this paper are listed below. As can be seen, ProbTS already accommodates a wide range of data scales, both in terms of the number of timesteps (ranging from 792 to 69,680) and the number of variates (ranging from 7 to 2,000). Please note that most existing studies only employ a subset of these datasets to evaluate the effectiveness of their neural architecture designs or probabilistic estimation strategies.
>
> | Dataset | # Var. | Timesteps |
> | :--- | :---: | ---: |
> | ETTh1-L / ETTh1-L | 7 | 17, 420 |
> | ETTm1-L / ETTm2-L | 7 | 69, 680 |
> | Electricity-L | 321 | 26, 304 |
> | Traffic-L | 862 | 17, 544 |
> | Weather-L | 21 | 52, 696 |
> | Exchange-L | 8 | 7, 588 |
> | ILI-L | 7 | 966 |
> | Exchange-S | 8 | 6, 071 |
> | Solar-S | 137 | 7, 009 |
> | Electricity-S | 370 | 5, 833 |
> | Traffic-S | 963 | 4, 001 |
> | Wikipedia-S | 2000 | 792 |
>
>
> **Response to Question 1**
>
> Thank you for bringing up these forward-thinking questions. We concur that zero-shot/few-shot/transfer/pretraining are all valuable and under-explored paradigms in the realm of time-series forecasting.
>
> As stated in our paper, our focus is on unifying probabilistic and non-probabilistic methods and facilitating comprehensive evaluations across diverse data characteristics. In doing so, we highlight many overlooked challenges, unresolved issues, and the strengths and weaknesses of existing approaches. Hence, we chose to concentrate on these aspects within this paper.
>
> Looking ahead, we believe that revealing these unique insights to the community will also be instrumental in advancing research in time-series forecasting within the zero-shot/few-shot/transfer/pre-training paradigms. This is because all these learning paradigms involve the selection of probabilistic or non-probabilistic estimations, custom or general architectures, and autoregressive or non-autoregressive decoding schemes. As for ProbTS itself, we believe it is certainly feasible to extend its scope beyond the supervised paradigm of time-series forecasting. This is due to the fact that we have compiled a wide range of datasets and aim to unify neural architecture designs and probabilistic estimation abilities. Based on your suggestion, we will include a discussion on these aspects in the revised paper, thus presenting a broader vision and greater potential impacts.
>
>
> **Response to Question 2**
>
> Currently, we have not encountered issues with RAM insufficiency. To facilitate machine learning on large quantities of data that do not fit into RAM, it's essential to establish a comprehensive data pipeline that systematically manages data sharding, indexing, shuffling, queuing, and so forth. For this purpose, we could learn from successful toolkits in language and vision domains. However, it's important to note that the focus of this paper is on unifying different research branches in time-series forecasting, rather than designing efficient pipelines to handle large-scale time-series data.
>
> Regarding your query about licenses, rest assured that all datasets utilized in our experiments are open-sourced and publicly accessible. We've compiled the sources of these datasets in the table below for your reference.
>
> | Dataset | Source |
> | :--- | :--- |
> | ETT | Zhou, Haoyi, et al. "Informer: Beyond efficient transformer for long sequence time-series forecasting." Proceedings of the AAAI. Vol. 35. No. 12. 2021. |
> | Electricity | https://archive.ics.uci.edu/ml/datasets/ElectricityLoadDiagrams20112014 |
> | Traffic | http://pems.dot.ca.gov/ |
> | Weather | https://www.bgc-jena.mpg.de/wetter/ |
> | Exchange | Lai, Guokun, et al. "Modeling long-and short-term temporal patterns with deep neural networks." Proceedings of SIGIR. 2018. |
> | ILI | https://gis.cdc.gov/grasp/fluview/fluportaldashboard.html |
> | Solar | https://forecastingdata.org/ |
> | Wikipedia | Gasthaus, Jan, et al. "Probabilistic forecasting with spline quantile function RNNs." The 22nd international conference on artificial intelligence and statistics. PMLR, 2019. |

---

### Author Response · Authors · 2023-11-23
**Paper Revision**

We extend our sincere gratitude once again to the reviewers for their perceptive comments.

In response to your suggestions, we have undertaken a revision of our paper. The revision incorporates: (a) emphasis on our unique contributions, (b) inclusion of additional experimental results to fortify the empirical evidence, (c) more comprehensive details about the model settings, (d) a discussion on the impact of data scale, and (e) miscellaneous clarifications.

We greatly appreciate all the questions and suggestions raised, as they have substantially improved the clarity and comprehensiveness of our paper's positioning and presentation.

---

**(a) Emphasis on our unique contributions**

[Reviewers AnhU and kFUJ]:

- **New Section 4.1** ("Insights Uncovered with ProbTS: An Overview"): Highlight our distinct findings, their relevance to unresolved challenges, and exciting future research opportunities.

**(b) Inclusion of additional experimental results to fortify the empirical evidence**

[Reviewers eUu7 and AnhU]

- **New Appendix E.2** ("Statistical and Gradient Boosting Decision Tree Baselines"): Added two statistical baselines (ARIMA and ETS) and a GBDT model (XGBoost) for a more comprehensive comparison.

[Reviewer eUu7]

- **New Appendix E.3** ("Experiments on Univariate Datasets"): Provided results on M4, M5, and TOURISM datasets to cover univariate and multivariate scenarios.

[Reviewer JfDV]

- **New Appendix E.4** ("Experiments on Synthetic Datasets"): Included experiments on synthetic datasets to enhance the rigor of presented insights.

**(c) More comprehensive details about the model settings**

[Reviewers AnhU and aD4Z]

- **New Appendix D.2** ("Hyper-parameters"): Included the key hyperparameter settings to offer readers a more accessible reference point.

**(d) Discussions on the impact of data scale**

[Reviewers AnhU, JfDV, and aD4Z]:

- **New Appendix E.1** ("Impact of Data Scale"): Attached a quantitative analysis of the relationship between data scale and method superiority.

**(e) Miscellaneous clarifications**

[Reviewer aD4Z]:

- **New Appendix F** ("Further Discussion on Cross-channel Interactions"): Included a discussion on cross-channel interactions in time series forecasting.

[Reviewer JfDV]:

- **Mention before definition**: We have now revised this issue.

[Reviewer kFUJ]:

- **Changed the "Model" column to "Studies" in Table 1**: We have now revised this issue.

---

With our clarifications and revisions, we trust that we have addressed the reviewers' concerns. We appreciate your consideration and look forward to your response.

---

### Meta-Review · Area_Chair_3CXr · 2023-12-06

**Metareview:**

The paper introduces a deep learning framework for time series models that combines and evaluates probabilistic and non-probabilistic methods. There was considerable disagreement between the reviewers in terms of the merits of the paper. It's strengths were generally acknowledged as (1) the introduction of a unified toolkit for times series forecasting; (2) analysis of short-term and long-term probabilistic and non-probabilistic models under different settings; (3) insights on the performance of various models on different time series datasets, and how the data characteristics relate to model performance. The weaknesses pointed out were (1) the lack of simple/common baselines, which was addressed; (2) the scale of the benchmarks, which was partially addressed, while the authors argued that diversity is more important than scale, an argument with which I mostly agree; (3) insufficient delving into the insights and lack of a rigorous analysis; (4) limited benefits compared to existing frameworks and insufficient comparisons against them. In their response to Reviewer kFUJ, the authors did provide a table of existing benchmarks, showing that, compared to TSLib, their framework is capable of probabilistic forecasting and compared to GluonTS, it offers long-term benchmarking and advanced neural network architectures. Reviewer kFUJ contested some of these claims about GluonTS.

Given that consensus was not reached, I read the paper myself. While I think it is well written, and would be a valuable resource to the community, my job is to make an adjudication of whether it is suited for the research track of a top tier ML conference. Here, unfortunately, I found it falls somewhat short. The architecture isn't groundbreaking, the insights are purely empirical, and, while there is some attempt at generalization, the field of time series analysis has hundreds, if not thousands of datasets that can be used. From the table presented, it looks like, between them Gluon TS and TSLib have the needed characteristics covered. Of course, to do side-by-side comparisons using these tools would require that they use some of the same datasets, but that seems like something that can be remedied. Also, some of the insights aren't very surprising, for instance "Modeling high dimensional probabilistic distributions demands substantial memory and is time-intensive". All in all, this paper is probably better suited to something like the datasets and benchmarks track at NeurIPS.

**Justification For Why Not Higher Score:**

Not all the reviewers were convinced by the paper. While the authors did address some of the issues raised by the reviewers, valid arguments were made about the limited technical contributions concerning the framework itself and the insights derived from it.

**Justification For Why Not Lower Score:**

N/A

---

### Decision · Program_Chairs · 2024-01-16

Reject